# iLoRA: Bayesian Low-Rank Adaptation with Latent Interaction Graphs for Microbiome Diagnosis

Yang Song [1]   Yixuan Zhang [1]   Lingfa Meng [1]   Tongyuan Hu [2]   Haizhou Shi [3]   Hao Wang [3]   Samir Bhatt [1 4]   Hengguan Huang [1 4]

## Abstract

Parameter-efficient adaptation has made LLMs practical for domain prediction, but standard LoRA still relies on a static low-rank update and does not expose the latent interactions that often drive scientific labels. We introduce iLoRA. To our knowledge, it is the first **Bayesian graph-conditioned LoRA framework**. It infers a **latent interaction graph** from the input and uses it to generate input-conditioned LoRA updates. As a result, iLoRA learns prediction and latent interaction structure jointly, rather than training a predictor and applying interaction analysis only post hoc. We instantiate this idea for microbiome diagnosis, where disease state can depend on both species-level abundance and microbe–microbe cross-talk, and evaluate it in two complementary settings: interactive QA with human-annotated graphs, which tests latent structure recovery, and multi-cohort IBD diagnosis, which tests biomedical utility. Across both settings, iLoRA improves over strong LoRA and Bayesian adaptation baselines, recovers graphs aligned with human annotations and cohort-level microbiome associations, and provides calibrated uncertainty with moderate graph-branch overhead.

[1] Section of Health Data Science & AI, Department of Public Health, University of Copenhagen, Copenhagen, Denmark [2] University of Copenhagen, Copenhagen, Denmark [3] Rutgers University, New Brunswick, NJ, USA [4] MRC Centre for Global Infectious Disease Analysis, Department of Infectious Disease Epidemiology, School of Public Health, Faculty of Medicine, Imperial College London, London, United Kingdom. Correspondence to: Hengguan Huang <hengguan.huang@sund.ku.dk>, Hao Wang <hw488@cs.rutgers.edu>, Samir Bhatt <samir.bhatt@sund.ku.dk>.

*Proceedings of the $43^{rd}$ International Conference on Machine Learning*, Seoul, South Korea. PMLR 306, 2026. Copyright 2026 by the author(s).

## 1. Introduction

Reliable microbiome-based diagnosis is becoming a practical requirement for precision medicine at scale, particularly for chronic inflammatory diseases such as inflammatory bowel disease (IBD). Yet the gut microbiome is not merely a collection of independent features: it is an ecosystem whose function and dysregulation emerge from *interactions* among microbial taxa (Faust & Raes, 2012). In IBD, disease signals manifest not only as species-level abundance shifts but also as changes in community organization and cross-talk, reflected in distorted co-occurrence topology and reorganized modular structure (Baldassano & Bassett, 2016) and in network-based biomarker discoveries (Hu et al., 2023). These observations motivate diagnostic models that can exploit interaction structure rather than treating taxa as exchangeable, conditionally independent covariates.

A parallel trend is the increasing use of large language models (LLMs) and post-training adaptation pipelines for domain-specialized prediction and decision support. Parameter-efficient adaptation methods such as Low-Rank Adaptation (LoRA) enable strong performance with minimal trainable parameters (Hu et al., 2022), making them attractive for deployment in biomedical settings where data and compute are constrained. However, standard post-training pipelines typically focus on improving predictive accuracy while overlooking *structured latent factors* that encode domain-specific dependencies. In microbiome applications, this gap is consequential: the clinically relevant signal may be distributed across coordinated taxa groups and interaction patterns, not isolated to marginal abundance changes. Moreover, clinical deployment requires not only accuracy but also calibrated uncertainty and robustness. Bayesian approximations such as dropout-based inference and ensemble-based uncertainty estimation have become standard tools for improving reliability in deep models (Gal & Ghahramani, 2016; Lakshminarayanan et al., 2017), and recent work has begun to bring Bayesian perspectives into low-rank adaptation itself (Yang et al., 2024; Wang et al., 2024).

Despite the biological importance of microbial interactions, inferring microbiome networks from abundance profiles

remains methodologically fragile. Microbiome measurements are compositional and sparse, and correlation- or association-based network estimates can vary substantially across methods and preprocessing choices; indeed, correlation detection strategies have been shown to differ widely in sensitivity and precision (Weiss et al., 2016). Downstream, disease studies often rely on post hoc pipelines that first train a predictor and then separately compute association networks, which can blur the statistical distinction between predictive features and mechanistic interactions. Complementary approaches—such as multivariable association discovery with covariate adjustment (Mallick et al., 2021) or generative simulators for benchmarking network inference (Qian et al., 2024)—help contextualize findings, but they do not, by themselves, yield an end-to-end *interaction-aware* diagnostic model.

We propose **iLoRA**. To our knowledge, it is the first **Bayesian graph-conditioned LoRA framework**. It infers a **latent interaction graph** from the input and uses it to generate input-conditioned LoRA updates. In microbiome diagnosis, the graph is therefore a predictive mechanism, not merely a post-hoc visualization: it directly conditions how the LLM is adapted for each sample. The Bayesian formulation supports principled uncertainty quantification over predictive outputs, bridging reliable diagnosis with ecosystem-level structure.

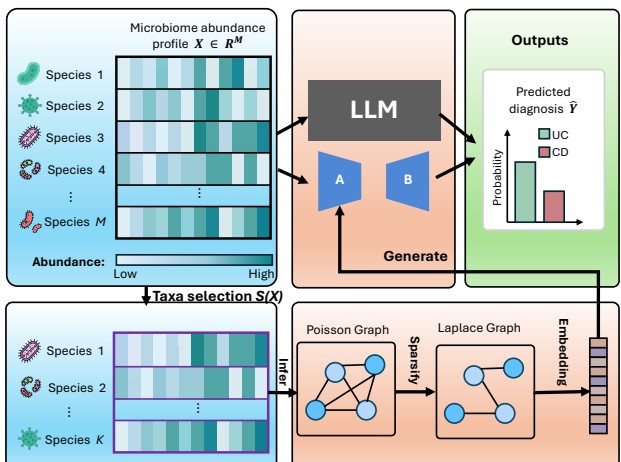

*Figure 1.* **Framework overview of iLoRA.** Given a microbiome abundance profile $X \in \mathbb{R}^M$, our model predicts diagnosis with an LLM while, in parallel, selecting $K < M$ key taxa to infer a latent interaction graph. We first infer a Poisson edge graph, then transform it into a sparse graph with Laplace-distributed edge weights to encourage sparsity, embed the graph with a GNN, and use the embedding to generate the LoRA matrix $A$. The outputs are the predicted diagnosis $\hat{Y}$ and a sparse interaction graph.

We evaluate iLoRA in two complementary settings. First, we use interactive question answering (QA) with human-annotated interaction graphs as a controlled benchmark for latent structure recovery, so Molweni is not merely an NLP

leaderboard but a test of whether the graph branch recovers meaningful relations while improving task performance (Li et al., 2020). Second, we evaluate iLoRA on gut microbiome cohorts for IBD diagnosis, comparing against strong LoRA baselines (Hu et al., 2022) and examining whether inferred graphs are consistent with conventional microbiome association networks and covariate-adjusted association analyses (Weiss et al., 2016; Mallick et al., 2021). Across both domains, iLoRA is designed to improve predictive accuracy while yielding interaction graphs that are interpretable and aligned with external relational evidence.

In summary, our contributions are:

- We introduce **iLoRA**[1]. To our knowledge, it is the first **Bayesian graph-conditioned LoRA framework**. It infers a **latent interaction graph** from the input and uses it to generate input-conditioned LoRA updates.

- iLoRA learns **prediction and latent interaction structure jointly**, rather than using interaction analysis only post hoc. In microbiome diagnosis, this turns microbe–microbe cross-talk into a sample-conditioned adaptation signal under a single end-to-end Bayesian objective, with Poisson edge modeling and Laplace-weighted sparsification providing uncertainty-aware sparse graph learning.

- We validate iLoRA in both **interactive QA with human-annotated graphs** and **real-world IBD diagnosis**, showing predictive gains and meaningful recovered interaction structure across language and biomedical settings.

## 2. Related Work

A related line of research can be broadly viewed as *mechanistic Bayesian reasoning*: methods that move beyond black-box prediction by discovering or explicitly modeling latent biological mechanisms from biomedical and biological data. Representative examples include semi-mechanistic Bayesian renewal-process models that infer latent transmission processes from noisy surveillance data (Bhatt et al., 2023), and Bayesian phylodynamic models that reconstruct latent viral transmission dynamics from pathogen genomic surveillance data (Khurana et al., 2024). Recent interaction- and dynamics-based mechanistic reasoning models further provide methodological primitives for such mechanism discovery, including BayesAgent for agentic graphical model reasoning (Huang et al., 2026), latent event-relational dynamics for EEG-based neurodegenerative classification (Feng et al., 2026), continuous-time Bayesian dynamics for non-stationary adaptation (Huang et al., 2022), stochastic

---

[1] https://github.com/GoodGoodMaul/iLoRA

boundary ordinary differential equations for learning unannotated event timing and dynamics (Huang et al., 2021), deep graph random processes for latent relational graph inference (Huang et al., 2020), and recurrent Poisson process units for modeling event-count dynamics in sequential data (Huang et al., 2019). Our work follows this mechanistic-AI perspective in microbiome diagnosis by treating microbe–microbe cross-talk as a latent, uncertainty-aware biological mechanism learned jointly with prediction, rather than as a post-hoc correlation artifact. We therefore organize the remaining related work around the two components that our method brings together: microbiome-based diagnosis and microbial interaction networks, followed by parameter-efficient adaptation with LLMs.

### 2.1. Microbiome-Based Diagnosis and Microbial Interaction Networks

Microbiome-based diagnosis typically learns predictors directly from abundance tables (or their engineered summaries), using statistical association testing and downstream classifiers. For IBD, recent diagnostic studies and community benchmarks demonstrate strong signal but also highlight practical challenges such as cohort shift, sparsity, and compositional effects that can degrade generalization across studies (Kang et al., 2023; Khachatryan et al., 2023; Mallick et al., 2021). Most diagnosis pipelines primarily treat taxa as independent covariates, so microbial interactions are incorporated only *implicitly* (e.g., through black-box predictors) or *post-hoc* (e.g., building a network after training to interpret discovered taxa).

In parallel, microbial interaction networks have been widely used to interpret dysbiosis beyond marginal abundances (Faust & Raes, 2012; Grilli et al., 2017). Network analyses in IBD report altered topology and reorganization of connectivity patterns in key taxa (Baldassano & Bassett, 2016; Hu et al., 2023), motivating interaction-aware modeling. However, *inferring* networks from sequencing data is nontrivial: correlation-based methods are sensitive to compositionality and zeros (Weiss et al., 2016), and even more robust approaches are commonly applied as separate preprocessing steps that output deterministic graphs without principled uncertainty and with limited support for biological constraints (Kurtz et al., 2015). By contrast, our approach couples *uncertainty-aware* sparse interaction discovery with diagnosis: we learn a probabilistic interaction structure and use it to condition the downstream decision model, rather than treating networks as a post-hoc explanatory artifact.

### 2.2. Parameter-Efficient Adaptation with LLMs

Parameter-efficient fine-tuning (PEFT) adapts large pretrained models with a small number of trainable parameters, including adapters (Houlsby et al., 2019), prompt/prefix tuning (Li & Liang, 2021; Lester et al., 2021), sparse parameter updates such as BitFit (Ben Zaken et al., 2022) and $IA^3$ (Liu et al., 2022), and low-rank weight updates such as LoRA (Hu et al., 2022). These methods are attractive for microbiome diagnosis because tabular datasets are often limited in size, and full fine-tuning can be unstable and computationally expensive (Mangrulkar et al., 2022). LoRA and its variants (e.g., quantized LoRA) provide strong compute–accuracy tradeoffs by learning low-rank updates on selected linear projections (Dettmers et al., 2023), but the learned update is typically *global*—shared across all samples—and does not explicitly encode relational structure in the input.

Our work targets this gap by making PEFT *structure-aware*: instead of relying solely on a static low-rank update (our LoRA baseline), iLoRA infers a latent graph for each input and uses that graph to generate the LoRA update itself. This distinguishes iLoRA from both static LoRA variants and post-hoc graph analyses: prediction and structure are learned together, and the recovered graph directly controls the adapter. This retains the efficiency and modularity of PEFT while enabling *sample-conditioned* adaptation driven by explicit microbial interactions, providing a principled bridge between interaction modeling and LLM-based diagnosis.

## 3. Background: Low-Rank Adaptation (LoRA)

Large language models are typically fine-tuned by updating all weights, which is costly and can overfit when supervision is limited. LoRA addresses this by freezing pretrained weights and learning a low-rank update for selected linear layers (Hu et al., 2022). Consider a linear transformation with weight matrix $W_0 \in \mathbb{R}^{d_{out} \times d_{in}}$. Instead of learning a full update $\Delta W$, LoRA parameterizes

$$W = W_0 + \Delta W, \qquad \Delta W = s\,BA, \qquad (1)$$

where $A \in \mathbb{R}^{r \times d_{in}}$ and $B \in \mathbb{R}^{d_{out} \times r}$ with rank $r \ll \min(d_{in}, d_{out})$, and $s$ is a scaling factor (often $s = \alpha/r$ in implementations) (Hu et al., 2022). During training, only $(A, B)$ are updated via standard backpropagation while $W_0$ remains fixed; gradients flow through $BA$ exactly as in a regular linear layer. A common initialization sets $B = \mathbf{0}$ and initializes $A$ with a small random matrix (e.g., Gaussian), so that $\Delta W = 0$ at the start and the model initially matches the pretrained network (Hu et al., 2022). At inference, the low-rank update can be merged into $W_0$ (i.e., use $W_0 + sBA$) without adding extra latency.

## 4. Problem Formulation

We consider supervised learning with *latent interaction graphs*; as a concrete example, in microbiome-based diagnosis the interactions among taxa are typically unob-

served. Given a dataset $\mathcal{D} = \{(X^{(n)}, y^{(n)})\}_{n=1}^N$, each input $X^{(n)} \in \mathbb{R}^M$ is a microbial abundance profile over $M$ observed taxa and $y^{(n)} \in \{0, 1\}$ indicates disease status. Since $M$ can be large in practice, we first apply a (possibly data-driven) filter/selection function $S(\cdot)$ to identify $K < M$ key taxa, yielding $Z^{(n)} = S(X^{(n)}) \in \mathbb{R}^K$ and a node set $V = \{1, \ldots, K\}$. For every sample $n$, we associate an unknown interaction graph $G^{(n)} = (V, E^{(n)})$ represented by an adjacency matrix $A^{(n)} \in [0, 1]^{K \times K}$, where $A_{ij}^{(n)} = 0$ denotes no edge and larger values indicate stronger co-occurrence (with $A_{ij}^{(n)} = 1$ as maximal co-occurrence); optionally we impose $A_{ii}^{(n)} = 0$ and symmetry $A^{(n)} = A^{(n)\top}$ for undirected interactions. The learning task is: given a new microbiome profile $X$, output both (i) a diagnosis prediction $\hat{y} \in \{0, 1\}$ and (ii) an inferred latent interaction graph $\hat{A} \in [0, 1]^{K \times K}$ over the selected $K$ taxa that annotates pairwise co-occurrence structure for that sample.

## 5. Method

### 5.1. Overview

Given a microbiome abundance profile $X \in \mathbb{R}^M$, our framework (Fig. 1) is inspired by the hierarchical Bayesian deep learning framework (Huang et al., 2020; Wang & Yeung, 2016; 2020; Wang et al., 2024) and follows a two-branch pipeline that couples *Bayesian inference of latent interaction graphs* with *LoRA-based LLM adaptation*. The key design choice is that the graph is not produced after prediction: it is the conditioning signal that generates the input-specific LoRA update.

In the **prediction branch**, we construct a lightweight prompt that encodes the taxa abundances in $X$ and feed it into a pretrained LLM to obtain a diagnosis prediction $\hat{y}$. In parallel, the **iLoRA branch** first applies a taxa selection function $S(\cdot)$ to extract $K < M$ key taxa, yielding $Z = S(X) \in \mathbb{R}^K$, and infers a sample-specific interaction graph over these taxa. Specifically, we first infer a *Poisson interaction graph* in which each edge variable is modeled with a Poisson distribution, capturing uncertain co-occurrence strength. To encourage sparsity, we then infer a sparse graph with Laplace-distributed edge weights by probabilistically transforming the Poisson edge variables into Laplace-distributed edge variables, yielding a sparse interaction graph $\hat{A} \in [0, 1]^{K \times K}$. Finally, we embed $\hat{A}$ using a graph neural network (GNN) and use the resulting graph representation to generate the LoRA update matrix $A$, producing a Bayesian graph-conditioned low-rank adaptation of the LLM. The model outputs both the predicted diagnosis $\hat{y}$ and the inferred sparse interaction graph $\hat{A}$ as an estimation of microbe–microbe co-occurrence structure.

### 5.2. Inferring a Poisson Graph from Multi-Entity Data

We represent multi-entity observations by a *Poisson interaction graph*: for a sample with $K$ selected entities (e.g., taxa in a microbiome profile), we introduce a latent graph whose edge $(i, j)$ is a nonnegative random variable $\tilde{\alpha}_{ij} \sim \text{Pois}(m_{ij})$, where $m_{ij}$ quantifies the (sample-specific) strength of co-occurrence between entities $i$ and $j$. In microbiome-based diagnosis, this provides a compact probabilistic abstraction of uncertain microbe–microbe interactions beyond marginal abundances, and serves as the latent structure inferred by the iLoRA branch.

**Node representations and edge-wise latent variables.** Given an input profile $X \in \mathbb{R}^M$, we first select $K < M$ key taxa via $Z = S(X)$, and denote the selected taxa names by $\{t_i\}_{i=1}^K$ with corresponding (normalized) abundances $\{z_i\}_{i=1}^K$. We obtain a node representation for each selected taxon by prompting the (frozen) LLM to encode the pair *(taxon name, abundance)*:

$$h_i = \text{Enc}_{\text{LLM}}(t_i, z_i) \in \mathbb{R}^d, \qquad i = 1, \ldots, K,$$

where $\text{Enc}_{\text{LLM}}(\cdot)$ denotes the resulting embedding (e.g., a hidden state from the LLM). For each unordered pair $(i, j)$, we then construct an edge feature vector

$$e_{ij} = \text{MLP}_\phi([h_i; h_j; |h_i - h_j|; h_i \odot h_j]),$$

which parameterizes the edge-wise latent variable $\tilde{\alpha}_{ij}$ in the Poisson interaction graph (and later its sparsity-inducing transformation). Collecting $\{\tilde{\alpha}_{ij}\}$ for all $i < j$ yields a sample-specific latent interaction structure over the selected $K$ taxa.

**Learning: ELBO for Poisson Graph.** Let $\tilde{\alpha} = \{\tilde{\alpha}_{ij}\}_{i<j}$ denote Poisson edge variables for the $K$ selected taxa. In the iLoRA branch, we learn a variational posterior $q_\varphi(\tilde{\alpha} \mid Z)$ by maximizing the evidence lower bound (ELBO):

$$\begin{aligned}
\log p_\theta(y \mid X) \geq &\ \mathbb{E}_{q_\varphi(\tilde{\alpha}|Z)}[\log p_\theta(y \mid X, \tilde{\alpha})] \\
&- \text{KL}(q_\varphi(\tilde{\alpha} \mid Z) \| p(\tilde{\alpha} \mid Z)).
\end{aligned} \tag{2}$$

Directly using a Poisson variational family is inconvenient for end-to-end learning because Poisson samples are discrete and do not admit a standard low-variance reparameterization, making gradient-based optimization and Monte Carlo estimation of the first term in (2) intractable. To obtain a differentiable pathwise estimator while retaining a Poisson interpretation, we introduce a *Gaussian proxy* for Poisson edges.

**Theorem 5.1** (Gaussian proxy for Poisson and closed-form rate matching). *Fix an edge $(i, j)$. Let the variational approximation be Gaussian,*

$$q_\varphi(\tilde{\alpha}_{ij} \mid Z) \triangleq \mathcal{N}(u_{ij}, \delta_{ij}^2), \qquad \delta_{ij} > 0, \tag{3}$$

*and use the Gaussian proxy for a Poisson random variable with rate $m > 0$, $\mathcal{N}(m, m)$. Define the matched Poisson rate $m_{ij}$ as the minimizer*

$$m_{ij} \triangleq \arg\min_{m>0} \mathrm{KL}\Big(\mathcal{N}(m, m) \,\|\, \mathcal{N}(u_{ij}, \delta_{ij}^2)\Big). \quad (4)$$

*Then the unique positive minimizer is*

$$m_{ij} = \frac{2u_{ij} - 1 + \sqrt{(2u_{ij} - 1)^2 + 8\delta_{ij}^2}}{4} > 0. \quad (5)$$

The proof is provided in the appendix. In practice, we sample the Gaussian variational edge as $\tilde{\alpha}_{ij} = u_{ij} + \delta_{ij}\epsilon_{ij}$ with $\epsilon_{ij} \sim \mathcal{N}(0, 1)$ and use the matched rate $m_{ij}$ in the Poisson-rate KL term. This preserves differentiable pathwise training while retaining the Poisson interpretation of edge intensity.

**Poisson prior and KL term.** We place an input-dependent Poisson prior on each edge,

$$p(\tilde{\alpha}_{ij} \mid Z) = \mathrm{Pois}\Big(m_{ij}^{(0)}\Big),$$
$$m_{ij}^{(0)} = \mathrm{Softplus}(f_{\phi_0}(e_{ij})). \quad (6)$$

where $f_{\phi_0}$ is a lightweight network (in the spirit of amortized priors used in latent-sequence models such as VRNN(Chung et al., 2015) parameterizations). With the Poisson rate $m_{ij}$ recovered from (5), the KL regularizer becomes the closed-form Poisson–Poisson divergence:

$$\mathrm{KL}\Big(\mathrm{Pois}(m_{ij}) \,\|\, \mathrm{Pois}(m_{ij}^{(0)})\Big) = m_{ij}^{(0)} - m_{ij}$$
$$+ m_{ij} \log \frac{m_{ij}}{m_{ij}^{(0)}}. \quad (7)$$

Summing (7) over all $i < j$ yields the graph regularization term in the ELBO (2).

### 5.3. From Poisson Edges to a Sparse Laplace-Weighted Graph

The Poisson interaction graph captures *nonnegative* co-occurrence strengths, but in many scientific graphs we also desire *sparsity*—most pairs of taxa should have negligible interaction. To obtain a sparse and signed interaction graph, we transform the Poisson-edge latent variables into *Laplace-distributed* edge variables, which impose a sharp peak at zero and heavy tails.

**Target sparse edge distribution.** For each unordered pair $(i, j)$, we introduce a sparse edge weight

$$\bar{\alpha}_{ij} \sim \mathrm{Laplace}(0, b_{ij}), \quad (8)$$

where the scale $b_{ij} > 0$ controls sparsity (smaller $b_{ij}$ encourages stronger shrinkage toward 0). Collecting $\{\bar{\alpha}_{ij}\}$ yields the Laplace-weighted sparse interaction graph used by the downstream GNN/LoRA modules.

**NPN transform: Poisson $\rightarrow$ Laplace.** Inspired by natural-parameter networks (NPNs) (Wang et al., 2016), we adopt neural networks to infer a target distribution from an input distribution through a sequence of probabilistic, sampling-free transformations. Concretely, for each edge $(i, j)$ we start from the inferred Poisson edge variable

$$\tilde{\alpha}_{ij} \sim \mathrm{Pois}(m_{ij}), \quad (9)$$

and apply an NPN mapping that outputs the parameters of a Laplace edge distribution:

$$(\mu_{ij}, b_{ij}) = \mathcal{T}_{\mathrm{NPN}}(\tilde{\alpha}_{ij}, e_{ij}),$$
$$\bar{\alpha}_{ij} \sim \mathrm{Laplace}(\mu_{ij}, b_{ij}). \quad (10)$$

In our implementation we set $\mu_{ij} = 0$ to center edges at zero and interpret $b_{ij}$ as a learned, edge-specific sparsity level. The role of the NPN is to propagate uncertainty from the Poisson graph into a Laplace family *without requiring discrete sampling* from $\mathrm{Pois}(m_{ij})$ during training.

**Scale-mixing view enabling efficient inference.** We exploit the standard Gaussian scale-mixture representation of the Laplace distribution: a Laplace random variable can be obtained by a Gaussian whose variance is randomly scaled. Specifically, for $\bar{\alpha}_{ij} \sim \mathrm{Laplace}(0, b_{ij})$, one convenient construction is

$$\bar{\alpha}_{ij} \mid \sigma_{ij}^2 \sim \mathcal{N}(0, \sigma_{ij}^2), \qquad \sigma_{ij} \sim \mathrm{Rayleigh}(b_{ij}), \quad (11)$$

which yields the marginal $\bar{\alpha}_{ij} \sim \mathrm{Laplace}(0, b_{ij})$. This representation allows the NPN to operate on continuous natural parameters (e.g., mapping uncertainty in $\tilde{\alpha}_{ij}$ to a distribution over $\sigma_{ij}$ and hence $b_{ij}$), while keeping training compatible with Gaussian-based backpropagation used in the Poisson stage.

### 5.4. Bayesian-calibrated prediction

The iLoRA branch yields a *posterior* over sparse interaction graphs through Laplace edge variables, which naturally supports uncertainty-aware prediction. Let $\bar{A}$ denote the Laplace-weighted interaction graph and let $p_\varphi(\bar{A} \mid X)$ be its (amortized) posterior implied by our Poisson$\rightarrow$Laplace construction. We define the Bayesian predictive distribution by marginalizing graph uncertainty:

$$p(y \mid X) = \mathbb{E}_{\bar{A} \sim p_\varphi(\bar{A}\mid X)}\big[p_\theta(y \mid X, \bar{A})\big]. \quad (12)$$

In practice we approximate (12) with Monte Carlo samples $\{\bar{A}^{(s)}\}_{s=1}^S$ drawn from the Laplace-weighted graph poste-

rior,

$$\hat{p}(y \mid X) \;=\; \frac{1}{S}\sum_{s=1}^{S} p_{\theta}\Big(y \mid X, \bar{A}^{(s)}\Big), \qquad (13)$$

which averages predictions from input-conditioned LoRA adaptations induced by different plausible graphs.

### 5.5. Learning objective

We train iLoRA end-to-end with variational inference:

$$
\begin{aligned}
\mathcal{L} = \;& \mathcal{L}_{\text{pred}} \\
& + \lambda_{\text{Pois}} \sum_{i<j} \text{KL}\Big(\text{Pois}(m_{ij}) \,\big\|\, \text{Pois}(m_{ij}^{(0)})\Big) \\
& + \lambda_{\text{Lap}} \sum_{i<j} \text{KL}(\text{Lap}(0, b_{ij}) \,\|\, \text{Lap}(0, b_0)).
\end{aligned} \qquad (14)
$$

where $\mathcal{L}_{\text{pred}}$ can be cross-entropy (or token NLL for a next-token classifier), and $b_0$ is a fixed prior Laplace scale. For zero-mean Laplace distributions, the KL reduces to

$$\text{KL}\Big(\text{Lap}(0, b_{ij}) \,\|\, \text{Lap}(0, b_0)\Big) = \log \frac{b_0}{b_{ij}} + \frac{b_{ij}}{b_0} - 1, \; (15)$$

and the Poisson KL is given in Eq. (7).

## 6. Experiments

We empirically validate iLoRA on two datasets designed to test distinct aspects of our framework: structural inference and diagnostic accuracy. First, we employ the Molweni dataset (Li et al., 2020) as a controlled benchmark for latent graph recovery, because its human-annotated discourse dependencies provide an external structural target rather than only an NLP score. Second, we apply our method to a large-scale, heterogeneous IBD microbiome cohort to evaluate its utility in precision medicine.

### 6.1. Experimental Setup

#### 6.1.1. DATASETS

**Multiparty Dialogue (Molweni).** To evaluate structural recovery, we utilize the Molweni dataset, which consists of multiparty dialogues annotated with discourse dependency structures. We treat each utterance as a node, allowing the model to infer latent graphs representing discourse relationships while performing machine reading comprehension. Thus, Molweni evaluates whether iLoRA recovers meaningful latent relations jointly with QA, not merely whether it improves a leaderboard metric.

**IBD Diagnosis.** We aggregated microbiome profiles from multiple publicly available independent cohorts to form a unified heterogeneous dataset. Key sources include Ananthakrishnan_2017, Franzosa_2019, and Lloyd-Price_2019, among others. A comprehensive list of all included cohorts, along with their specific bibliographic references, is provided in Appendix E. The unified microbiome representation consists of 3,061 species-level microbial taxa as input features. After restricting to patients with consistent UC/CD labels, the binary diagnosis subset contains 1,014 patient samples, each represented by these species-level features. We focus on the binary classification of **Ulcerative Colitis (UC) vs. Crohn's Disease (CD)**. To ensure rigorous evaluation, the dataset was partitioned into training, validation, and test sets with a ratio of approximately $7 : 1.5 : 1.5$ (710 training samples, 152 validation, 152 test). Crucially, we performed *stratified splitting at the cohort level* to guarantee that the test split contains samples from diverse study populations, testing the model's generalization capability.

*Table 1.* Results on Molweni (Span Extraction). Best results are in **bold**.

| Method | F1 | EM |
|---|---|---|
| Zero-shot | 56.32 | 35.70 |
| MLE | 72.83 | 57.78 |
| MAP | 72.66 | 57.02 |
| BLOB | 70.80 | 55.90 |
| MCD | 72.33 | 57.51 |
| ENS | 72.38 | 57.09 |
| **iLoRA (Ours)** | **74.51** | **60.57** |

Feature selection was performed using **MaAsLin2** (Mallick et al., 2021), identifying the top 20 significant species based on FDR-adjusted $q$-values (detailed feature list in Appendix Table 10).

#### 6.1.2. TASK FORMULATION

We formulate the learning objectives as follows:

- **Molweni (Span Extraction):** We treat the task as autoregressive generation. The model is instructed to extract the minimal text span answering a query and output it in a structured JSON format.

- **IBD Diagnosis (Next-Token Prediction):** We reformulate the classification as a next-token prediction problem. The LLM acts as a domain expert, receiving the full list of non-zero microbial species and prompted to generate exactly one token: "yes" (UC) or "no" (CD).

The exact prompt templates for both tasks are provided in Table 12 of the Appendix. Detailed definitions of evaluation metrics and checkpoint selection criteria are provided in Appendix D.

*Table 2.* Relation-prediction error rates for the structural benchmarks.

| Benchmark | Graph Type | Error Rate |
|---|---|---|
| Molweni | Random Graph | 50.0 |
| Molweni | **Inferred Graph (iLoRA)** | **26.7** |
| IBD | Random Graph | 50.0 |
| IBD | **Inferred Graph (iLoRA)** | **27.3** |

### 6.1.3. BASELINES

We compare iLoRA against standard PEFT methods (**MLE**, **MAP**) and state-of-the-art uncertainty-aware adaptations. Specifically, we evaluate **Monte-Carlo Dropout (MCD)** (Gal & Ghahramani, 2016), which approximates the posterior via dropout during inference; **Deep Ensemble (ENS)** (Lakshminarayanan et al., 2017; Balabanov & Linander, 2025; Wang et al., 2023), which aggregates predictions from multiple independently trained LoRA adapters; **BLOB** (Wang et al., 2024), a variational inference approach using backpropagation for Bayesian LoRA; and, where applicable, **Laplace LoRA (LAP)** (Yang et al., 2024), which applies a post-hoc Laplace approximation to the parameter posterior. LAP is evaluated only for the IBD next-token diagnosis task. We do not report LAP on Molweni because LAP is specifically designed for closed-ended prediction rather than autoregressive span generation; see Appendix C for details.

**Implementation Details.** We implement the iLoRA module as a graph hypernetwork where a structural graph encoder dynamically generates the LoRA $A$ matrix based on the latent interaction graph, while the $B$ matrix remains static across samples. Detailed neural architectures, tensor shapes, and the complete training protocol are provided in Appendix B and C.

### 6.2. Experiment I: Multiparty Dialogue and Structure Recovery

We first evaluate the model's performance on the Molweni dataset, focusing on both the downstream span extraction task and the structural quality of the learned latent graphs. We report these two outcomes jointly because the central claim is not only higher QA accuracy, but prediction that is coupled to a recoverable latent interaction structure.

### 6.2.1. QUANTITATIVE RESULTS: DOWNSTREAM ACCURACY AND STRUCTURAL RECOVERY

Table 1 summarizes the performance on the machine reading comprehension task. iLoRA achieves a state-of-the-art F1 score of **74.51%** and an Exact Match (EM) score of **60.57%**.

**Discussion.** iLoRA significantly outperforms both the standard fine-tuning baselines (MLE, MAP) and uncertainty-aware methods. In multiparty dialogue, the correct answer often hinges on the interaction history between specific speakers rather than the linear text sequence. This pattern is consistent with the intended role of the latent graph: it helps the model emphasize structurally relevant utterances and reduce reliance on irrelevant conversational context. Notably, iLoRA surpasses Deep Ensemble (ENS) without the computational overhead of maintaining multiple models, validating the efficiency of our latent graph approach. Read together with Table 2, these gains show that iLoRA improves QA while reducing graph error from 50.0% to 26.7%, supporting Molweni as a controlled structure-recovery benchmark.

**Structural Error Analysis.** To explicitly validate the structural recovery capabilities of iLoRA, we measured the *Error Rate* of the inferred adjacency matrices compared to the ground-truth connectivity. As shown in Table 2, a random graph baseline yields an error rate of **50.0%**. In contrast, our proposed iLoRA framework achieves a significantly lower error rate of **26.7%**. This substantial reduction indicates that **iLoRA** successfully captures meaningful discourse dependencies rather than relying on random associations.

### 6.2.2. QUALITATIVE CASE STUDY: VISUALIZATION OF LATENT GRAPHS

To intuitively demonstrate the properties of the learned structures, we visualize the inferred interaction graphs for two representative samples. Detailed dialogue content for these cases is provided in **Appendix H**.

**Disentangling Interleaved Conversations.** Figure 2 (**top**) shows the inferred interaction graph for Sample 1371. The dialogue contains two semantically distinct threads: one discussing software compilation (U0–U2) and another regarding hardware driver issues (U3–U7). The inferred graph closely matches the block-diagonal structure, effectively separating the two threads. Specifically, in terms of *Cluster Identification*, the model predicts high connectivity within the first group (U0–U2) and the second group (U3–U7), mirroring the ground truth. Regarding *Noise Suppression*, the model predicts zero interactions between the unrelated threads (e.g., U0 vs. U4), suggesting that iLoRA selectively utilizes the latent graph to focus on relevant context.

**Recovering Discourse Chains.** Figure 2 (**bottom**) illustrates Sample 2568, a troubleshooting session characterized by a sequence of clarifications. iLoRA successfully recovers key discourse dependencies. For *Question–Answer Identification*, the model correctly predicts the strong interaction between the question in U5 and the answer in U6. Furthermore, regarding *Contextual Flow*, the graph aligns with the chain of interaction from U3 to U4 and subsequently to U5, matching the logical flow of the troubleshooting steps.

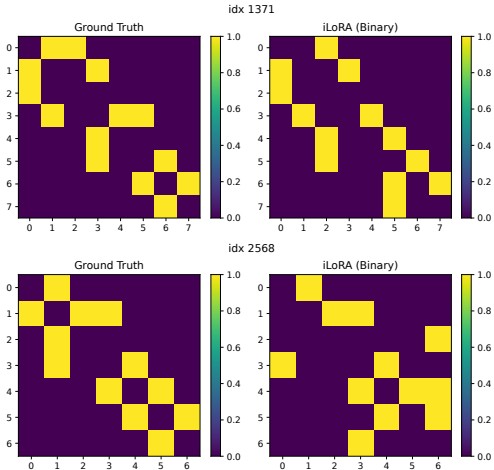

*Figure 2.* Visualization of latent interaction graphs. Left: Ground-truth adjacency matrix. Right: Inferred graph by iLoRA. The top example (Sample 1371) shows thread disentanglement, while the bottom (Sample 2568) shows discourse chain recovery.

*Table 3.* Overall performance on IBD diagnosis (UC vs. CD). Best metrics are highlighted in **bold**.

| Method | ECE ↓ | F1 (UC) ↑ | AUROC ↑ | AUPRC ↑ |
|---|---|---|---|---|
| MLE | 0.2533 | 0.6071 | 0.7617 | 0.7570 |
| MAP | 0.2082 | 0.6496 | 0.7637 | 0.7117 |
| MCD | 0.2762 | 0.6341 | 0.7428 | 0.7117 |
| ENS | 0.1598 | 0.5794 | 0.7574 | 0.7565 |
| BLOB | 0.1570 | 0.5882 | 0.7812 | 0.7577 |
| LAP | 0.2031 | 0.6496 | 0.7641 | 0.7122 |
| **iLoRA (Ours)** | **0.0980** | **0.6557** | **0.7990** | **0.7617** |

## 6.3. Experiment II: IBD Diagnosis and Interaction Discovery

### 6.3.1. DIAGNOSTIC PERFORMANCE AND CALIBRATION

Table 3 presents the aggregated results for the UC vs. CD classification task. iLoRA demonstrates superior diagnostic capability, achieving the highest **AUROC (0.7990)** and **AUPRC (0.7617)** among all evaluated methods. These metrics underscore the model's robustness in distinguishing between disease subtypes and its effectiveness in maintaining high precision across varying decision thresholds, which is essential for minimizing false positives in clinical screening.

**Discussion.** iLoRA shows three complementary advantages. *Discriminative power:* it surpasses strong uncertainty-aware baselines in ranking performance, outperforming BLOB (0.7812) and Deep Ensemble (0.7574) in AUROC, which suggests that latent microbial interactions provide a richer signal for separating complex disease phenotypes than strictly weight-space ensembles. *Calibration:* iLoRA achieves an ECE of **0.0980**, substantially lower than MLE (0.2533) and MAP (0.2082), producing probability estimates better aligned with empirical accuracy. *Class balance:* iLoRA also obtains the highest UC F1-score (0.6557), indi-

*Table 4.* Ablation study on iLoRA. The iLoRA (w/o Laplace) keeps the latent Poisson graph branch but removes Laplace sparsification.

| Variant | ECE ↓ | F1 (UC) ↑ | AUROC ↑ | AUPRC ↑ |
|---|---|---|---|---|
| MLE (vanilla LoRA) | 0.2533 | 0.6071 | 0.7617 | 0.7570 |
| iLoRA (w/o Laplace) | 0.1032 | 0.6341 | 0.7557 | 0.7440 |
| iLoRA (full) | **0.0980** | **0.6557** | **0.7990** | **0.7617** |

*Table 5.* Comparison with standard tabular baselines on IBD diagnosis using the same selected taxa.

| Model | F1 (UC) ↑ | AUROC ↑ | AUPRC ↑ |
|---|---|---|---|
| Random Forest | 0.5753 | 0.6151 | 0.6467 |
| XGBoost | 0.5292 | 0.5823 | 0.6467 |
| MLP | 0.4906 | 0.5346 | 0.6214 |
| iLoRA | **0.6557** | **0.7990** | **0.7617** |

cating that the gain is not limited to threshold-free ranking but also improves subtype identification at the operating point. The result supports the graph-conditioned design: the inferred interaction structure is used to adapt the model, rather than only to explain it after training.

We further validate robustness by analyzing performance across eight independent cohorts (see Appendix Table 15). iLoRA shows strong generalization, particularly on major cohorts like *Franzosa_2019B* (AUROC 0.9500) and *Lee_2021* (AUROC 0.8611).

**Component ablation.** We next examine which components of the latent graph branch contribute to the IBD diagnosis results. Table 4 compares vanilla LoRA, a Poisson-only variant that removes Laplace sparsification, and the full iLoRA model. The Poisson graph substantially improves calibration over vanilla LoRA, reducing ECE from 0.2533 to 0.1032. Adding Laplace sparsification further improves discriminative performance, increasing AUROC from 0.7557 to 0.7990 and AUPRC from 0.7440 to 0.7617. This suggests that the Poisson stage helps represent uncertainty in latent interactions, while the Laplace sparsity prior suppresses noisy edges and yields a more task-relevant interaction structure. Vanilla LoRA serves as the no-graph-conditioned reference. We therefore ablate components that can be removed while preserving graph-conditioned adaptation; removing the GNN/hypernetwork graph-to-LoRA interface would collapse the method toward a qualitatively different non-graph-conditioned adapter rather than isolate one optional component.

**Comparison with tabular baselines.** Because the IBD input is ultimately derived from microbial abundance features, we also compare against standard non-LLM tabular baselines trained on the same selected taxa. As shown in Table 5, iLoRA outperforms Random Forest (Breiman, 2001), XGBoost (Chen & Guestrin, 2016), and MLP (Rumelhart et al., 1986) baselines by a large margin in AUROC and

AUPRC. These results indicate that the gains are not simply an artifact of the selected feature set, but arise from the interaction-aware adaptation mechanism.

### 6.3.2. STATISTICAL TAXA ASSOCIATION REFERENCE

In this section, we construct a *model-agnostic cohort-level statistical reference* using the same $n = 152$ samples and $p = 20$ taxa as iLoRA, to characterize taxon–taxon relationships supported by standard cohort-level statistics and to assess how iLoRA's *sample-conditioned* interaction graphs agree with and complement these global patterns. This is a structural reference for learned edges; predictive comparisons are reported separately against LoRA, Bayesian adaptation, and tabular baselines. We define significant taxon-pair sets from two perspectives: (i) microbe–microbe co-variation in a compositionally corrected space, capturing stable co-occurrence or mutual exclusion patterns; and (ii) log-ratio pairwise features tested against the diagnostic label $y$ (CD vs. UC), yielding phenotype-associated contrasts. Pooling across these two perspectives yields a cohort-level reference set of 41 significant taxon pairs. These reference sets serve as an external baseline to quantify enrichment of iLoRA's high-weight edges and to interpret additional structure beyond cohort-level marginal tests.

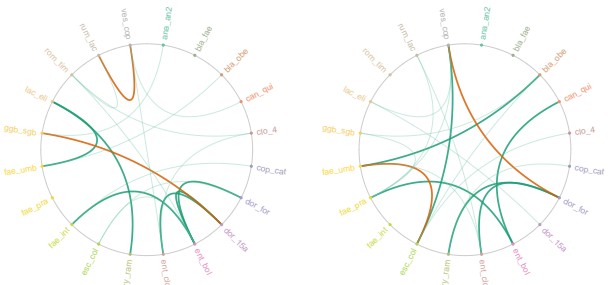

*(a)* pub_S402 | Kumbhari_2024   *(b)* pub_S114 | Kumbhari_2024

*Figure 3.* Chord diagram visualizations for two representative samples from the Kumbhari_2024 cohort.

For iLoRA's sample-level graphs, we symmetrize directed scores and select the top $10\%$ of edges per sample ($K = 19$) ranked by descending weight. We then evaluate these edges against the global significant set of 41 taxon pairs. As shown in Table 2, iLoRA substantially outperforms a random-graph baseline (expected error rate 50.0%), achieving an error rate of 27.3%. This indicates that iLoRA's strongest inferred interactions are globally concentrated on cohort-supported associations rather than being randomly distributed. The complete results and analysis details in the Appendix J.

Furthermore, high-weight non-overlapping edges in the iLoRA graphs can capture additional structural information beyond cohort-level marginal statistical screening. Across two representative samples (pub_S402 and pub_S114; Fig. 3), we highlight in green six edges that

overlap with the 41 globally significant taxon pairs, and in orange several high-weight pairs that are not detected by the cohort-level screening but exhibit clear biological relevance in the sample-level graphs, including *E. coli–F. umbilicata*, *D. formicigenerans–V. coprocola*, *Dorea sp. AF36-15AT–GGB9453_SGB14844*, and *R. lactaris–V. coprocola*. These sample-specific edges are consistent with prior IBD evidence: *E. coli* has been reported as enriched among CD biomarkers, whereas *D. formicigenerans* tends to be depleted; *V. coprocola* has been linked to inflammatory readouts including fecal calprotectin; and *R. lactaris* appears protective in multi-omic IBD risk models (Zheng et al., 2024; Gorman & Lladser, 2024). Moreover, given the established bile-acid–microbiome axis in IBD, associations involving a Dorea-lineage taxon and an uncharacterized SGB provide a biologically plausible context for sample-specific network rewiring (Bai et al., 2024; Duboc et al., 2013). Taken together, the IBD experiments provide a compact three-way check on the same mechanism: prediction improves over LoRA and Bayesian baselines, calibration improves substantially, and high-weight edges are enriched in independently supported taxon pairs. Thus, the learned graph is useful as a conditioning signal for diagnosis, not merely as a visualization of the fitted model. Across domains, Molweni tests recovery against human-annotated discourse graphs, whereas IBD tests the same mechanism under sparse species-level features and a cohort-level statistical reference. This pairing is intentionally complementary: one benchmark makes the latent structure observable, while the other asks whether sample-conditioned structure improves a realistic biomedical prediction task.

## 7. Conclusion

We presented iLoRA. To our knowledge, it is the first Bayesian graph-conditioned LoRA framework: it infers a latent interaction graph from the input and uses it to generate input-conditioned LoRA updates. Given an input profile, iLoRA infers a Poisson interaction graph, transforms it into a sparse graph with Laplace-distributed edge weights, and embeds the resulting latent structure with a GNN to generate per-example LoRA updates while keeping the LLM backbone frozen. This design delivers better predictions together with an explicit interaction structure, and further enables Bayesian-calibrated prediction by marginalizing over graph uncertainty via Monte Carlo estimation. Empirically, iLoRA consistently improves over standard PEFT and SoTA Bayesian adaptation baselines for both language and biomedical settings, while simultaneously recovering meaningful structure. Looking forward, the graph-to-adaptation principle can be extended beyond microbiomes to other multi-entity domains (e.g., multi-omics). We further discuss limitations regarding feature selection, graph scalability, inference overhead, task scope, and causal interpretation in Appendix K.

## Impact Statement

This work contributes to microbiome-based precision medicine by enabling joint, uncertainty-aware clinical prediction and discovery of latent microbial interaction structure, addressing a key limitation of correlation-based post hoc analyses. By integrating Bayesian, parameter-efficient adaptation with end-to-end learning of interaction graphs, the proposed framework improves diagnostic performance while yielding interpretable representations of microbe-microbe cross-talk, with direct relevance to inflammatory bowel disease and broader applicability to structured modeling in complex biological systems.

## Acknowledgments

We thank all reviewers, SPC, and AC for their valuable comments. S.B. acknowledges support from the Novo Nordisk Foundation via The Novo Nordisk Young Investigator Award (NNF20OC0059309). S.B. acknowledges support from The Eric and Wendy Schmidt Fund For Strategic Innovation via the Schmidt Polymath Award (G-22-63345) which also supports HH and LM. S.B. acknowledges support from the Novo Nordisk Foundation via the Global Pathogen Analysis Platform (GPAP) (NNF26SA0109818). HW is supported by Amazon Faculty Research Award, Microsoft AI & Society Fellowship, NSF CAREER Award IIS-2340125, NIH grant R01CA297832, and NSF grant IIS-2127918.

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

## A. Proof of Theorem 5.1

*Proof.* Fix an edge $(i, j)$ and write $u \triangleq u_{ij}$ and $\delta^2 \triangleq \delta_{ij}^2$. For $m > 0$, define

$$P_m \triangleq \mathcal{N}(m, m), \qquad Q \triangleq \mathcal{N}(u, \delta^2).$$

Using the closed form KL divergence between univariate Gaussians,

$$\mathrm{KL}\big(\mathcal{N}(\mu_0, \sigma_0^2) \,\|\, \mathcal{N}(\mu_1, \sigma_1^2)\big) = \frac{1}{2} \left( \log \frac{\sigma_1^2}{\sigma_0^2} + \frac{\sigma_0^2 + (\mu_0 - \mu_1)^2}{\sigma_1^2} - 1 \right),$$

we obtain the objective

$$
\begin{aligned}
f(m) &\triangleq \mathrm{KL}(P_m \| Q) \\
&= \frac{1}{2} \left( \log \frac{\delta^2}{m} + \frac{m + (m - u)^2}{\delta^2} - 1 \right), \qquad m > 0.
\end{aligned}
\tag{16}
$$

This function is strictly convex on $(0, \infty)$ because

$$f''(m) = \frac{1}{2} \left( \frac{1}{m^2} + \frac{2}{\delta^2} \right) > 0.$$

Hence the minimizer is unique and is characterized by $f'(m) = 0$. Differentiating (16) gives

$$f'(m) = \frac{1}{2} \left( -\frac{1}{m} + \frac{1 + 2(m - u)}{\delta^2} \right).$$

Setting $f'(m) = 0$ and rearranging yields

$$-\frac{1}{m} + \frac{1 + 2(m - u)}{\delta^2} = 0 \quad \Longleftrightarrow \quad m\big(1 + 2(m - u)\big) = \delta^2,$$

equivalently the quadratic equation

$$2m^2 + (1 - 2u)m - \delta^2 = 0.$$

Its discriminant is

$$\Delta = (1 - 2u)^2 + 8\delta^2 = (2u - 1)^2 + 8\delta^2 > 0,$$

so the solutions are

$$m = \frac{-(1 - 2u) \pm \sqrt{\Delta}}{4} = \frac{2u - 1 \pm \sqrt{(2u - 1)^2 + 8\delta^2}}{4}.$$

Because $\sqrt{\Delta} > |2u - 1|$ for $\delta^2 > 0$, the "+" root is strictly positive while the "−" root is negative. Therefore, the unique minimizer on $m > 0$ is

$$m^\star = \frac{2u - 1 + \sqrt{(2u - 1)^2 + 8\delta^2}}{4} > 0.$$

Finally, since the variational approximation is Gaussian, sampling $\tilde{\alpha}_{ij} = u + \delta \, \epsilon$ with $\epsilon \sim \mathcal{N}(0, 1)$ is reparameterizable and provides a pathwise estimator for expectations under $q_\varphi(\tilde{\alpha}_{ij} \mid Z)$. $\qquad\square$

## B. Architectures and Training Details

This section provides a detailed description of the components used in the iLoRA framework, the specific mechanisms for latent graph inference, and the training protocol utilized for both the Molweni and IBD diagnosis tasks.

## B.1. Input Processing and Embedding

**IBD Microbiome Encoding.**    Unlike standard text inputs, microbiome profiles are inherently tabular and sparse. We adopt a feature-to-text transformation to bridge this gap. Let

$$X = \{(n_1, v_1), \ldots, (n_{20}, v_{20})\}$$

denote the sequence of the top 20 significant microbial feature pairs, where $n_i$ represents the paired microbe names and $v_i$ denotes their corresponding normalized abundances, as identified via MaAsLin2 (see Sec. F). Each feature pair is serialized into a natural-language description encoding both identity and abundance information.

To align these biological features with the semantic space of large language models, we obtain initial embeddings using the frozen embedding layer of a pre-trained Qwen3-8B model (Yang et al., 2025). Specifically, the resulting representation

$$H^{(0)} \in \mathbb{R}^{B \times N \times D_m}$$

serves as the initial node embedding tensor, where $B$ is the batch size, $N$=20 is the sequence length corresponding to the number of feature pairs, and $D_m$ denotes the hidden dimension of the LLM. These embeddings are subsequently used as input node features for the structural encoder.

**Molweni Dialogue Encoding.**    For the multiparty dialogue task, utterances are tokenized and encoded using the Llama-3.1-8B tokenizer (Touvron et al., 2023). The distinct speaker tokens and discourse markers are preserved to maintain conversational structure.

## B.2. Latent Graph Inference Module (iLoRA)

The core of our method is the extraction of a latent interaction graph $G = (V, E)$ from the input embeddings, which subsequently conditions the LoRA adaptation.

**Node and Edge Representation.**    The input embeddings $X$ are first projected to a lower-dimensional graph space via a linear layer, yielding node embeddings $H \in \mathbb{R}^{N \times d_g}$ (where $d_g = 128$). To capture pairwise dependencies, we construct an edge embedding tensor $E_{ij}$ by concatenating node pairs:

$$E_{ij} = \text{MLP}_{\text{edge}}([h_i \parallel h_j]), \quad E_{ij} \in \mathbb{R}^{2d_g} \tag{17}$$

This pairwise representation forms the basis for our Bayesian edge inference.

**NPN-based Variational Inference.**    To model the uncertainty and sparsity of interactions, we utilize NPN to parameterize the edge distributions. We employ a two-branch encoder to estimate the prior and posterior distributions of edge weights:

- **Prior Network:** Estimates a Poisson rate parameter $m_{ij}^{(0)}$ representing the prior expected interaction strength.

- **Posterior Network:** Estimates the mean $\mu_{ij}$ and variance $\sigma_{ij}^2$ of a Gaussian approximation to the Poisson edge-count distribution.

To induce sparsity, we perform a mapping from the Gaussian posterior to a **Laplace** distribution via the CDF transformation technique. The final edge weight $A_{ij}$ is sampled via reparameterization:

$$A_{ij} = \text{ReLU}(\text{Sample}_{\text{Laplace}}(\mu_{ij}, \sigma_{ij}^2)) \tag{18}$$

This ensures that the inferred graph is sparse and non-negative, interpretable as interaction strengths.

**Graph Convolution and Matching Attention.**    The sampled adjacency matrix $A$ is used to propagate information via a 2-layer Graph Convolutional Network (GCN):

$$H^{(l+1)} = \sigma(D^{-1/2} \tilde{A} D^{-1/2} H^{(l)} W^{(l)}) \tag{19}$$

The structure-aware node embeddings are then fused with the original context via a **Matching Attention** mechanism, which computes an attention alignment between the graph-refined representations and the original sentence embeddings.

## B.3. Hypernetwork and LoRA Injection

Unlike standard LoRA, which learns static matrices $A$ and $B$, iLoRA functions as a hypernetwork. The pooled graph representation $h_{graph}$ is projected to generate the LoRA update weights dynamically for each input sample. Specifically, the hypernetwork generates the LoRA $A$ matrix for the final adapted layer of the LLM, while sharing a static $B$ matrix across samples.

$$\Delta W = s \cdot B \cdot \text{reshape}(\text{MLP}_{\text{proj}}(h_{graph})) \tag{20}$$

where $\text{MLP}_{\text{proj}} : \mathbb{R}^{d_g} \to \mathbb{R}^{r \times d_{in}}$ generates the input-conditioned $A$ matrix, while $B$ remains static or is jointly optimized. This allows the LLM to adapt its processing logic based on the inferred interaction structure of the current sample.

## B.4. Optimization and Protocol

**Loss Function.** The total objective function $\mathcal{L}$ is a weighted sum of the task-specific loss (Cross-Entropy) and Bayesian regularization terms for the latent graph:

$$\mathcal{L} = \mathcal{L}_{\text{CE}} + \lambda_{\text{Lap}}\text{KL}(Q_{\text{Lap}} \parallel P_{\text{Prior}}) + \lambda_{\text{Pois}}\text{KL}(Q_{\text{Pois}} \parallel P_{\text{Pois}}) \tag{21}$$

where $\lambda_{\text{Lap}}$ and $\lambda_{\text{Pois}}$ control the strength of the structural priors.

**Training Strategy.** We employ the AdamW optimizer. We freeze the LLM backbone and only update the LoRA adapters, the iLoRA graph module, and (for IBD) the class token embeddings.

**Evaluation.** For classification, we use the probability of the specific tokens ("yes"/"no") associated with the classes. For span extraction, we employ constrained beam search to generate valid JSON outputs.

## B.5. Shapes Summary

Table 6 details the tensor transformations through the iLoRA module.

*Table 6.* Tensor shapes in the iLoRA module (Configured for IBD Task).

| Module | Input Shape | Output Shape | Description |
|---|---|---|---|
| Input Embeddings | $\mathbb{R}^{B \times N \times 4096}$ | $\mathbb{R}^{B \times N \times 128}$ | Linear Projection to Graph Dim |
| Pairwise Constructor | $\mathbb{R}^{B \times N \times 128}$ | $\mathbb{R}^{B \times N^2 \times 256}$ | Concat node pairs for all $i, j$ |
| NPN Posterior (Mean/Var) | $\mathbb{R}^{B \times N^2 \times 256}$ | $\mathbb{R}^{B \times N^2 \times 1}$ | Estimate edge distribution parameters |
| Sampled Adjacency | Parameters | $\mathbb{R}^{B \times N \times N}$ | Sparse Adjacency Matrix $A$ |
| GCN Layer 1 | $\mathbb{R}^{B \times N \times 128}$ | $\mathbb{R}^{B \times N \times 128}$ | Graph Convolution |
| GCN Layer 2 | $\mathbb{R}^{B \times N \times 128}$ | $\mathbb{R}^{B \times N \times 128}$ | Graph Convolution |
| Readout Pooling | $\mathbb{R}^{B \times N \times 128}$ | $\mathbb{R}^{B \times 128}$ | Mean pooling over nodes |
| Hypernetwork Head | $\mathbb{R}^{B \times 128}$ | $\mathbb{R}^{B \times (r \cdot 4096)}$ | Generates LoRA $A$ weights |

## B.6. Pseudocode Summaries

---

**Algorithm 1** iLoRA Forward Pass (Latent Graph to Weights)

---

**Require:** Input embeddings $X \in \mathbb{R}^{N \times D}$, Masks $M$
**Ensure:** LoRA Adapter Weights $\Delta W$, KL losses
 1: **Step 1: Node & Edge Encoding**
 2: $H \leftarrow \text{Linear}(X)$ {Project to graph dim}
 3: $E_{ij} \leftarrow \text{Concat}(H_i, H_j)$ for all $i, j$
 4: **Step 2: Bayesian Edge Inference (NPN)**
 5: $\mu_{ij}, \sigma_{ij} \leftarrow \text{Encoder}_{\text{Post}}(E_{ij})$ {Posterior params}
 6: $m_{ij} \leftarrow \text{Encoder}_{\text{Prior}}(E_{ij})$ {Prior Poisson rate}
 7: $A_{ij} \leftarrow \text{SampleLaplace}(\mu_{ij}, \sigma_{ij})$ {Reparameterized sampling}
 8: $A \leftarrow \text{ReLU}(A)$ {Enforce non-negativity}
 9: **Step 3: Structure-Aware Encoding**
10: $H_{graph} \leftarrow \text{GCN}(H, A)$ {Message passing}
11: $H_{fused} \leftarrow \text{MatchingAttention}(H, H_{graph})$
12: $h_{ctx} \leftarrow \text{MeanPool}(H_{fused}, M)$
13: **Step 4: Weight Generation**
14: $W_{lora} \leftarrow \text{MLP}_{\text{hyper}}(h_{ctx})$
15: $\Delta W \leftarrow \text{Reshape}(W_{lora}, (r, D_{in}))$
16: **Step 5: Regularization**
17: $\mathcal{L}_{KL} \leftarrow \text{KL}_{\text{Laplace}} + \text{KL}_{\text{Poisson}}$
18:
19: **return** $\Delta W, \mathcal{L}_{KL}$

---

**Algorithm 2** NPN Sparse Edge Sampling

---

**Require:** Posterior Gaussian params $\mu, \sigma$
**Ensure:** Sparse edge weight $w$
 1: $\epsilon \sim \mathcal{N}(0, 1)$
 2: $z \leftarrow \mu + \sigma \cdot \epsilon$ {Gaussian proxy}
 3: $u \leftarrow \Phi(z)$ {Map to Uniform via Gaussian CDF}
 4: $w \leftarrow F_{\text{Laplace}}^{-1}(u)$ {Inverse Transform Sampling}
 5:
 6: **return** $w$

---

## C. Implementation Details

All experiments were implemented using the PyTorch framework and the Hugging Face PEFT library (Mangrulkar et al., 2022). Training was performed with mixed-precision (BF16 for IBD, FP16 for Molweni) to optimize computational efficiency.

**Model Architectures.** For the **Molweni** span extraction task, we utilized the **Llama-3.1-8B-Instruct** (Touvron et al., 2023) as the backbone. For the **IBD Diagnosis** task, we utilized **Qwen3-8B** (Yang et al., 2025).

- **LoRA Configuration:**
  - *Molweni:* We applied LoRA to projection layers (q_proj, k_proj, v_proj, o_proj) with rank $r = 8$, scaling factor $\alpha = 16$, and dropout 0.05.
  - *IBD Diagnosis:* We applied LoRA to the same projection layers but with rank $r = 16$ and $\alpha = 32$. Additionally, we set the embedding layer (embed_tokens) and language model head (lm_head) as trainable modules to accommodate the task-specific class tokens.

**Training Hyperparameters.** We utilized the **AdamW** optimizer for all experiments. Due to the differing nature of the tasks (span extraction vs. long-context binary classification), specific hyperparameters were tuned separately. These are detailed in Table 7.

*Table 7.* Hyperparameter settings for iLoRA experiments across tasks.

| Hyperparameter | Molweni (Span Extraction) | IBD Diagnosis |
|---|---|---|
| Batch Size | 4 | 2 |
| Learning Rate | $1 \times 10^{-4}$ | $2 \times 10^{-4}$ |
| Epochs | 3 | 6 |
| Warmup | Ratio 0.06 | Steps 100 |
| Weight Decay | 0.0 | 0.0 |
| Max Sequence Length | 1024 | 9000 |
| *Bayesian Regularization* | | |
| Laplace KL Weight ($\lambda_{\text{Lap}}$) | tuned in $[10^{-4}, 3 \times 10^{-3}]$ | tuned in $[10^{-4}, 3 \times 10^{-3}]$ |
| Poisson KL Weight ($\lambda_{\text{Pois}}$) | tuned in $[10^{-4}, 3 \times 10^{-3}]$ | tuned in $[10^{-4}, 3 \times 10^{-3}]$ |

*Table 8.* Inference cost comparison on the IBD diagnosis task. Latency is measured per sample.

| Model | Latency (ms/sample) $\downarrow$ | GPU memory (MB) $\downarrow$ |
|---|---|---|
| LoRA (MLE) | 377.2 | 21263.9 |
| ENS | 1144.6 | 21263.9 |
| iLoRA | 567.5 | 21330.3 |

**Hardware.** All models were fine-tuned on NVIDIA RTX 6000 Pro Blackwell PCIe GPUs and NVIDIA A100 80GB PCIe GPUs.

**Baseline Configurations. MAP baseline.** For MAP training, we set `weight_decay`=0.01 in AdamW, treating the resulting solution as a maximum-a-posteriori estimate under an $\ell_2$ Gaussian prior on trainable parameters.

**Uncertainty Baseline Configurations.** To ensure a fair comparison, baselines were configured as follows:

- **Monte-Carlo Dropout (MCD):** We used a dropout rate of $p = 0.05$ and performed 5 stochastic forward passes during inference to estimate the posterior predictive distribution.

- **Deep Ensemble (ENS):** We trained an ensemble of 3 independent LoRA adapters initialized with different random seeds. For classification, we averaged the logits; for text generation, we employed majority voting on the generated tokens.

- **BLOB:** We utilized the variational inference approach with $N = 10$ Bayesian evaluation samples during the forward pass.

- **Laplace LoRA (LAP):** We implemented a post-hoc Kronecker-factored Laplace approximation on pre-trained MAP checkpoints. We evaluate LAP on the IBD next-token diagnosis task, where prediction reduces to a fixed binary token decision ("yes"/"no") and the likelihood/curvature computation is directly comparable across PEFT baselines. We do not report LAP for Molweni span extraction. Extending this post-hoc LAP implementation to autoregressive span generation would require sequence-level or token-wise curvature estimation and uncertainty propagation through decoding, which is non-trivial and not directly comparable to the fixed-label IBD setting.

**Computational Overhead.** We measure inference latency and GPU memory on the IBD diagnosis setting under the same backbone and evaluation protocol. At inference time, iLoRA uses one graph sample ($S = 1$) by default. As shown in Table 8, iLoRA introduces moderate overhead compared with vanilla LoRA due to the sample-specific graph inference and hypernetwork generation, but remains substantially faster than Deep Ensemble. This overhead is expected because iLoRA performs pairwise edge inference over the selected taxa and then generates an input-conditioned LoRA update, whereas Deep Ensemble requires multiple independently trained adapters. Because the default setting uses $S = 1$, the added latency is dominated by a single graph-hypernetwork pass rather than repeated multi-sample adapter evaluation; larger $S$ can be used when stronger marginalization over graph uncertainty is desired.

## D. Evaluation Metrics and Model Selection

**IBD Diagnosis.** Distinguishing UC from CD based solely on microbiome profiles is a non-trivial challenge for conventional machine learning models (Kang et al., 2023). Motivated by this clinical difficulty, we focus specifically on the binary classification task of **UC vs. CD**.

To comprehensively assess performance in this imbalanced setting, we employ **Expected Calibration Error (ECE)**, **F1-score (UC class)**, **AUROC**, and **AUPRC** (Saito & Rehmsmeier, 2015) as evaluation metrics. We select the best model checkpoint based on the optimal **AUROC** achieved on the validation set.

**Multiparty Dialogue (Molweni).** For the span extraction task, we utilize standard machine reading comprehension metrics: **Exact Match (EM)** and **F1-score**. The best checkpoint was selected based on the highest **F1-score** on the validation set.

## E. Microbiome Dataset Details

To ensure the robustness and generalizability of iLoRA, we integrated microbiome profiles from distinct independent studies. These cohorts cover a diverse range of patient populations and sequencing methodologies. Table 9 lists the specific cohort identifiers used in our experiments alongside their corresponding primary academic citations.

*Table 9.* List of independent microbiome cohorts used in the aggregated IBD dataset and their corresponding references.

| Cohort ID | Study Reference | Citation |
|---|---|---|
| Ananthakrishnan_2017 | Ananthakrishnan et al. (2017) | (Ananthakrishnan et al., 2017) |
| Franzosa_2019 (B/N) | Franzosa et al. (2019) | (Franzosa et al., 2019) |
| Khachatryan_2023 | Khachatryan et al. (2023) | (Khachatryan et al., 2023) |
| Kumbhari_2024 | Kumbhari et al. (2024) | (Kumbhari et al., 2024) |
| Lee_2021 | Lee et al. (2021) | (Lee et al., 2021) |
| Lloyd-Price_2019 | Lloyd-Price et al. (2019) | (Lloyd-Price et al., 2019) |
| Ning_2023 | Ning et al. (2023) | (Ning et al., 2023) |

## F. Feature Selection (MaAsLin2)

Microbiome data is high-dimensional and sparse. We employed **MaAsLin2** (Mallick et al., 2021) (Microbiome Multivariable Associations with Linear Models) to identify significant microbial features distinguishing UC from CD. We selected the top 20 species based on FDR-adjusted $q$-values to serve as the structured nodes for our probabilistic graph inference module. Table 10 lists these features.

We further evaluate the sensitivity of iLoRA to the number of selected taxa $K$. For each value of $K$, we select the top-$K$ taxa according to MaAsLin2 FDR-adjusted $q$-values and keep the remaining training and evaluation protocol fixed. Table 11 shows that $K = 20$ provides the best overall operating point, achieving the best ECE, AUROC, and AUPRC. Increasing $K$ to 30 or 40 does not improve ranking performance and worsens calibration, suggesting that additional low-signal taxa mainly introduce noisy candidate edges in the $O(K^2)$ graph branch.

## G. Prompt Formulations

Table 12 outlines the exact templates used for both tasks. For the IBD diagnosis task, the "Microbiome Profile" section of the prompt is dynamically populated with the non-zero microbial species from each specific sample, sorted by relative abundance.

## H. Case Study Dialogue Content

Table 13 and Table 14 present the raw dialogue text for the two case studies discussed in the main text.

*Table 10.* Top 20 significant microbial species identified by MaAsLin2 for distinguishing UC from CD. These species act as nodes in the iLoRA latent interaction graph.

| Microbial Feature | Coef | StdErr | P-value | Q-value |
|---|---|---|---|---|
| *Faecalibacterium prausnitzii* | -3.261 | 0.418 | $1.51 \times 10^{-14}$ | $3.10 \times 10^{-12}$ |
| *Lachnospira eligens* | -3.524 | 0.459 | $3.78 \times 10^{-14}$ | $3.89 \times 10^{-12}$ |
| *Candidatus Cibionibacter quicibialis* | -2.850 | 0.424 | $3.09 \times 10^{-11}$ | $2.12 \times 10^{-9}$ |
| *Faecalimonas umbilicata* | 2.125 | 0.320 | $4.94 \times 10^{-11}$ | $2.12 \times 10^{-9}$ |
| *GGB9453 SGB14844* | -2.134 | 0.321 | $5.14 \times 10^{-11}$ | $2.12 \times 10^{-9}$ |
| *Dorea sp AF36 15AT* | -2.386 | 0.374 | $2.66 \times 10^{-10}$ | $9.13 \times 10^{-9}$ |
| *Enterocloster bolteae* | 2.047 | 0.324 | $4.11 \times 10^{-10}$ | $1.15 \times 10^{-8}$ |
| *Ruminococcus lactaris* | -1.863 | 0.296 | $4.46 \times 10^{-10}$ | $1.15 \times 10^{-8}$ |
| *Dorea formicigenerans* | -2.234 | 0.356 | $5.18 \times 10^{-10}$ | $1.19 \times 10^{-8}$ |
| *Blautia faecis* | -2.287 | 0.368 | $7.18 \times 10^{-10}$ | $1.48 \times 10^{-8}$ |
| *Vescimonas coprocola* | -2.106 | 0.344 | $1.27 \times 10^{-9}$ | $2.38 \times 10^{-8}$ |
| *Coprococcus catus* | -1.887 | 0.312 | $2.03 \times 10^{-9}$ | $3.21 \times 10^{-8}$ |
| *Enterocloster clostridioformis* | 2.040 | 0.337 | $1.90 \times 10^{-9}$ | $3.21 \times 10^{-8}$ |
| *Faecalibacillus intestinalis* | -2.260 | 0.378 | $3.03 \times 10^{-9}$ | $4.46 \times 10^{-8}$ |
| *Erysipelatoclostridium ramosum* | 2.532 | 0.426 | $3.79 \times 10^{-9}$ | $5.21 \times 10^{-8}$ |
| *Blautia obeum* | -2.378 | 0.412 | $1.02 \times 10^{-8}$ | $1.31 \times 10^{-7}$ |
| *Clostridium sp AF36 4* | -2.441 | 0.425 | $1.22 \times 10^{-8}$ | $1.47 \times 10^{-7}$ |
| *Romboutsia timonensis* | -1.874 | 0.330 | $1.69 \times 10^{-8}$ | $1.89 \times 10^{-7}$ |
| *Escherichia coli* | 2.508 | 0.441 | $1.74 \times 10^{-8}$ | $1.89 \times 10^{-7}$ |
| *Anaeromassilibacillus sp An250* | -1.647 | 0.290 | $1.86 \times 10^{-8}$ | $1.92 \times 10^{-7}$ |

*Table 11.* Sensitivity to the number of selected taxa $K$ on IBD diagnosis.

| $K$ | ECE $\downarrow$ | F1 (UC) $\uparrow$ | AUROC $\uparrow$ | AUPRC $\uparrow$ |
|---|---|---|---|---|
| 10 | 0.1082 | 0.6393 | 0.7545 | 0.7337 |
| 20 | **0.0980** | 0.6557 | **0.7990** | **0.7617** |
| 30 | 0.1217 | 0.5405 | 0.7574 | 0.6782 |
| 40 | 0.1438 | **0.6719** | 0.7359 | 0.6593 |

# I. Detailed Results per IBD Cohort

Table 15 provides the comprehensive performance breakdown for iLoRA and all baseline methods across the eight independent datasets used in the IBD diagnosis task.

| Dataset | Prompt Template and Example |
|---|---|
| **Molweni**

(Span Extraction) | **System Prompt:** You are a helpful, respectful and honest assistant. Always answer as helpfully as possible... [Safety Boilerplate]
**User Input:**
*[Task Instruction]* Extract the minimal span, word for word, from the following context that best answers the question. Output only the answer in the following JSON format...
*[Context]* sipher: bacon5o there 's no " fixmbr " with ubuntu ... [truncated] ...
*[Question]* Why does Bacon5o not want ubuntu ?

**Target Output:** {"answer":  "it does n't support my internet"} |
| **IBD Diagnosis**

(Next-Token) | **User Input:**
*[Role & Task Instruction]* You are a gastroenterologist focused on distinguishing Ulcerative Colitis (UC) from Crohn's Disease (CD). Based on the microbiome profile, answer exactly one token: 'yes' for UC or 'no' for CD.
*[Sample Info]*
 • Sample ID: CRR1110479 \| Ning_2023
 • Source dataset: updated_metaphlan202403_subtype_Ning_2023
 • Expected answers: UC → yes; CD → no

*[Microbiome Profile]* Detected non-zero microbial species and their relative abundances:
 - *Eisenbergiella_massiliensis*: 0.2666
 - *Escherichia_coli*: 0.1723
 ... [List of non-zero species sorted by abundance] ...
 - *Proteus_mirabilis*: 0.0000098
 Answer:

**Target Output:** no |

*Table 12.* Prompt formulations for the Molweni and IBD Diagnosis tasks.

## J. Details of the statistical analysis

We define cohort-level sets of significant taxon pairs from two complementary perspectives: on the one hand, we analyze microbe–microbe co-variation in a compositionally corrected space to obtain edges that reflect stable co-occurrence/exclusion structure; on the other hand, we test log-ratio–based pairwise features against the diagnostic label $y$ (CD vs. UC) to obtain edges that capture compositionally interpretable contrasts associated with the phenotype.

**(i) Cohort-level microbe–microbe association: CLR–Spearman + BH–FDR.** Let $x_{ij}$ denote the relative abundance of taxon $j$ in sample $i$. Because microbial abundances satisfy the closure constraint $\sum_j x_{ij} = 1$, computing correlations directly in the original proportion space is affected by well-known compositional artifacts. We therefore evaluate microbe–microbe association in a log-ratio space. Specifically, on the selected set of 20 taxa we add a small pseudocount $c > 0$, perform row-wise closure, and compute the centered log-ratio (CLR) transform:

$$\tilde{x}_{ij} = \frac{x_{ij} + c}{\sum_{\ell=1}^{p}(x_{i\ell} + c)}, \tag{22}$$

$$z_{ij} = \log \tilde{x}_{ij} - \frac{1}{p}\sum_{\ell=1}^{p} \log \tilde{x}_{i\ell}, \tag{23}$$

yielding $Z \in \mathbb{R}^{n \times p}$.

For each undirected taxon pair $(j, k)$ among the $\binom{20}{2} = 190$ candidates, we define the cohort-level association strength via the Spearman rank correlation

$$w_{jk}^{\text{Spear}} = \rho_s(Z_{\cdot j}, Z_{\cdot k}), \tag{24}$$

where $w_{jk}^{\text{Spear}} > 0$ indicates a co-occurrence trend and $w_{jk}^{\text{Spear}} < 0$ indicates a tendency toward mutual exclusion. We compute two-sided $p$-values for all pairs and apply Benjamini–Hochberg (BH) correction over the 190 tests to control the

*Table 13.* Dialogue content for Sample 1371 (Case Study 1). The conversation contains two distinct topics.

| ID | Speaker | Utterance |
|----|---------|-----------|
| | | ***Thread A: Building tar files*** |
| U0 | technodude | can you build a tar file on a debian based distro ? |
| U1 | cafuego | of course . just make sure you use .FILEPATH – prefix=FILEPATH |
| U2 | linux-rulz | yes , i do it all the time with things like mplayer and ffmpe |
| | | ***Thread B: Alcatel USB Driver Issues*** |
| U3 | predaeus | t 's a alcatel usb speedtouch |
| U4 | cafuego | probably a badly ported driver , then . |
| U5 | cafuego | can you swap it over for an ethernet one ? |
| U6 | predaeus | no should and must work with usb |
| U7 | cafuego | no , alcatel have provided a driver which does n't work under 64bit kernels , it seems . |

*Table 14.* Dialogue content for Sample 2568 (Case Study 2). The conversation involves troubleshooting a server mounting issue.

| ID | Speaker | Utterance |
|----|---------|-----------|
| U0 | mrwes | mount shares from ubuntu server FILEPATH ... you changed this to match your information ? |
| U1 | Doonz | nope didnt here is everything URL |
| U2 | mrwes | there are two in front of your ip |
| U3 | mrwes | and when you did a mkdir you used sharedfiles or server ? |
| U4 | Doonz | the extra must be something pastebin |
| U5 | mrwes | what fs is the share ? ntfs or ext3 ? |
| U6 | Doonz | i can connect to the share through places - connect to server |

false discovery rate (FDR), obtaining $q_{jk}^{\mathrm{Spear}}$. We then define the set of cohort-level significant microbe–microbe associations as

$$E_{\mathrm{Spear}} = \{(j,k) : q_{jk}^{\mathrm{Spear}} < 0.05\}.$$

Fig.4 provides an overview of $E_{\mathrm{Spear}}$, including the sign and FDR-adjusted strength of association to show the microbe–microbe association.

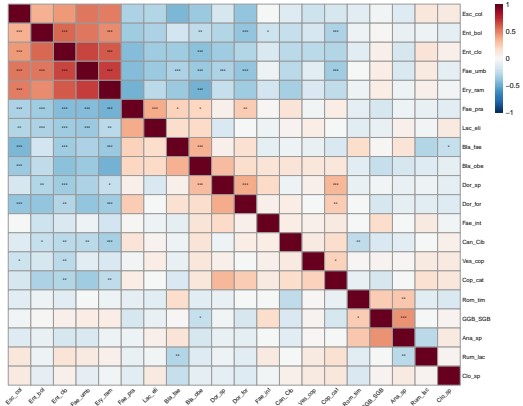

*Figure 4.* Heatmap of microbe–microbe association

**(ii) Cohort-level pair $\to y$ association: log-ratio + logistic + BH–FDR (Fig. 6b).** To characterize compositional contrasts associated with diagnosis, we construct, for each taxon pair $(a, b)$, a log-ratio feature

$$s_{ab,i} = \log(x_{ia} + \epsilon) - \log(x_{ib} + \epsilon), \tag{25}$$

*Table 15.* Full performance breakdown (ECE, F1-UC, AUROC, AUPRC) for iLoRA and all baselines across individual IBD cohorts. **MCD**: Monte-Carlo Dropout; **ENS**: Deep Ensemble; **LAP**: Laplace LoRA.

| Dataset Cohort | Method | ECE | F1 (UC) | AUROC | AUPRC |
|---|---|---|---|---|---|
| **Kumbhari_2024** | MLE | 0.2444 | 0.6341 | 0.7643 | 0.7621 |
| | MAP | 0.2641 | 0.6154 | 0.7507 | 0.6582 |
| | MCD | 0.3168 | 0.5778 | 0.7296 | 0.6674 |
| | ENS | 0.1107 | 0.6829 | 0.7894 | **0.8089** |
| | BLOB | 0.1449 | **0.6842** | **0.8207** | 0.8047 |
| | LAP | 0.2243 | 0.6154 | 0.7486 | 0.6563 |
| | **iLoRA** | **0.1029** | 0.6522 | 0.7976 | 0.7321 |
| **Lee_2021** | MLE | **0.1701** | **0.8000** | 0.8264 | **0.8717** |
| | MAP | 0.2723 | 0.7143 | 0.8194 | 0.8575 |
| | MCD | 0.2170 | **0.8000** | 0.7639 | 0.8374 |
| | ENS | 0.1768 | 0.6667 | 0.8333 | 0.8517 |
| | BLOB | 0.2163 | 0.6154 | 0.8472 | 0.8507 |
| | LAP | 0.2557 | 0.7143 | 0.8056 | 0.8534 |
| | **iLoRA** | 0.1738 | **0.8000** | **0.8611** | 0.8671 |
| **Khachatryan_2023** | MLE | 0.5202 | 0.2500 | 0.5000 | **0.6593** |
| | MAP | 0.4944 | 0.4444 | 0.5500 | 0.6117 |
| | MCD | **0.3571** | **0.7273** | **0.6500** | **0.7783** |
| | ENS | 0.4449 | 0.2500 | 0.4000 | 0.5311 |
| | BLOB | 0.3429 | 0.2857 | 0.4500 | 0.5726 |
| | LAP | 0.4870 | 0.4444 | 0.5500 | 0.6117 |
| | **iLoRA** | 0.4852 | 0.4000 | 0.3500 | 0.5060 |
| **Franzosa_2019B** | MLE | 0.1901 | 0.6667 | 0.8438 | 0.8849 |
| | MAP | 0.2098 | 0.6667 | 0.9375 | 0.9324 |
| | MCD | **0.1701** | **0.8571** | **0.9500** | **0.9472** |
| | ENS | 0.2361 | 0.7692 | 0.9375 | 0.9324 |
| | BLOB | 0.2300 | 0.5455 | 0.8375 | 0.8368 |
| | LAP | 0.2099 | 0.6667 | 0.9250 | 0.9249 |
| | **iLoRA** | 0.1794 | 0.7143 | **0.9500** | 0.9415 |
| **Ning_2023** | MLE | 0.3879 | 0.0000 | **0.6556** | 0.4153 |
| | MAP | 0.2918 | **0.3636** | 0.5333 | 0.3485 |
| | MCD | 0.3769 | 0.3333 | 0.6333 | 0.4817 |
| | ENS | 0.2818 | 0.2222 | 0.5333 | 0.4397 |
| | BLOB | 0.2600 | 0.0000 | 0.6000 | 0.3588 |
| | LAP | 0.2910 | **0.3636** | 0.5444 | 0.3624 |
| | **iLoRA** | **0.0964** | 0.2857 | **0.7278** | **0.6500** |
| **Lloyd-Price_2019** | MLE | 0.3133 | 0.5000 | **0.7143** | 0.7442 |
| | MAP | 0.2929 | 0.6000 | **0.7143** | 0.6976 |
| | MCD | 0.2257 | 0.5714 | **0.7143** | **0.7976** |
| | ENS | **0.1947** | 0.3333 | **0.7143** | 0.7076 |
| | BLOB | 0.2763 | **0.7500** | 0.6286 | 0.7633 |
| | LAP | 0.2909 | 0.6000 | **0.7143** | 0.6976 |
| | **iLoRA** | 0.3581 | 0.4444 | 0.6143 | 0.6176 |
| **Ananthakrishnan_2017** | MLE | 0.2926 | 0.6667 | 0.7381 | 0.8341 |
| | MAP | **0.0825** | **0.9231** | **0.8571** | **0.9341** |
| | MCD | 0.3764 | 0.5455 | 0.5952 | 0.7508 |
| | ENS | 0.4370 | 0.4000 | 0.7619 | 0.8044 |
| | BLOB | 0.3371 | 0.4444 | 0.8095 | 0.8600 |
| | LAP | 0.0836 | **0.9231** | **0.8571** | **0.9341** |
| | **iLoRA** | 0.1589 | 0.8333 | 0.8214 | 0.8984 |
| **Franzosa_2019N** | MLE | **0.1001** | **1.0000** | **1.0000** | **1.0000** |
| | MAP | 0.1760 | 0.8889 | **1.0000** | **1.0000** |
| | MCD | 0.3087 | 0.7500 | 0.5833 | 0.6458 |
| | ENS | 0.2427 | 0.7500 | 0.9167 | 0.9500 |
| | BLOB | 0.3285 | 0.8889 | 0.7500 | 0.8042 |
| | LAP | 0.1806 | 0.8889 | **1.0000** | **1.0000** |
| | **iLoRA** | 0.1671 | 0.8889 | 0.8333 | 0.8875 |

where $\epsilon > 0$ is used for zero stabilization. This construction is exactly equivalent to the difference of CLR coordinates,

$$s_{ab,i} = \text{CLR}(a)_i - \text{CLR}(b)_i, \tag{26}$$

and is therefore coherent within the compositional framework, with a direct interpretation: larger $s_{ab,i}$ indicates that taxon $a$ dominates taxon $b$ more strongly in sample $i$. Moreover, this "single-taxon vs. single-taxon" log-ratio can be viewed as the special case of a balance contrast where each side of the balance consists of a single taxon, in which case it reduces to $\log(x_A/x_B)$. This choice is consistent with balance/log-ratio–based association analyses widely used in clinical microbiome studies, for example in the longitudinal immune checkpoint blockade (ICB) melanoma cohort of Björk et al. (Björk et al., 2024), where taxon-level log-ratio summaries are related to treatment response and disease status.

We fix CD as the positive class ($y = 1$, UC $= 0$) and, for each pair, fit a univariate logistic regression model

$$\Pr(y_i = 1 \mid s_{ab,i}) = \sigma(\alpha + \beta_{ab}\, s_{ab,i}), \tag{27}$$

testing $H_0 : \beta_{ab} = 0$ to obtain $p_{ab}^{\text{ratio}}$. We then apply BH–FDR correction over the 190 tests to obtain $q_{ab}^{\text{ratio}}$ and define

$$E_{\text{ratio}} = \{(a,b) : q_{ab}^{\text{ratio}} < 0.05\}. \tag{28}$$

When (quasi-)complete separation leads to unstable maximum-likelihood estimates, we employ bias-reduced or penalized logistic regression to obtain stable coefficient estimates; this implementation detail does not alter the screening rule itself. With CD defined as the positive class, $\beta_{ab} > 0$ indicates that increasing $\log(A/B)$ is associated with higher log-odds of CD, while $\beta_{ab} < 0$ corresponds to the opposite direction. Fig.5 summarizes the direction of effects and FDR-supported significance across all pairs.

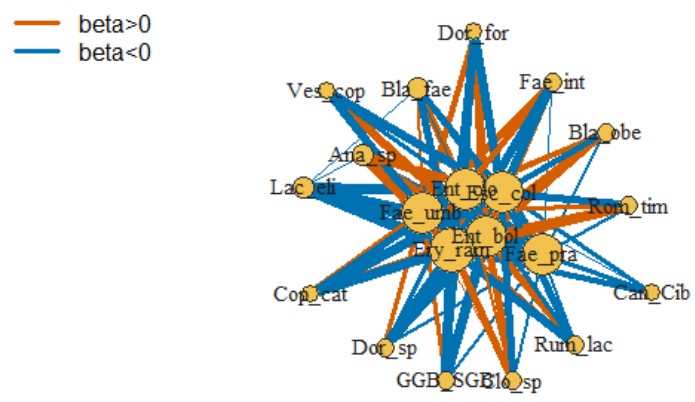

*Figure 5.* Pair-Y (CD vs UC): conditional log-ratio network (q¡0.05)

**(iii) Definition and role of the intersection reference set $E_{\text{GT}}$.** The two screening procedures yield edge sets $E_{\text{Spear}}$ and $E_{\text{ratio}}$, respectively. We define their intersection as the cohort-level statistical reference set:

$$E_{\text{GT}} = E_{\text{Spear}} \cap E_{\text{ratio}}. \tag{29}$$

In our dataset, $|E_{\text{GT}}| = 41$. At the chosen FDR threshold, this set collects taxon pairs that have *both* compositionality-corrected evidence of co-variation and log-ratio–based evidence of association with diagnosis. In the subsequent analysis,

we explicitly treat $E_{\mathrm{GT}}$ as a *cohort-level statistical reference*, rather than a complete "true network": through the use of an intersection and multiple-testing control, the construction is conservative with respect to false positives and may therefore miss interactions that occur only in specific subgroups of samples or that depend on higher-order conditional structures. Such potentially missed signals are precisely the type of structure that sample-conditioned modeling is designed to complement.

**(iv) Comparison with iLoRA sample-level graphs: overlap enrichment and sample-conditioned structure.** For each sample, iLoRA outputs directed edge strengths. To align with the undirected taxon pairs in the reference, we symmetrize the outputs for each sample $i$ as

$$\tilde{w}_{jk}^{(i)} = \frac{w_{j \to k}^{(i)} + w_{k \to j}^{(i)}}{2}. \tag{30}$$

Among all 190 undirected pairs, we then select, per sample, the top $10\%$ edges according to $\tilde{w}_{jk}^{(i)}$, i.e., $K = \lceil 0.1 \times 190 \rceil = 19$, yielding a predicted edge set $P_K^{(i)}$. Concretely, $P_K^{(i)}$ is obtained by ranking all candidate undirected pairs $(j, k)$ in descending order of $\tilde{w}_{jk}^{(i)}$ and taking the top $K$ edges. Equivalently, define the rank

$$r_{jk}^{(i)} = 1 + \sum_{(u,v) \in \mathcal{E}} \mathbb{I}\left( \tilde{w}_{uv}^{(i)} > \tilde{w}_{jk}^{(i)} \right), \qquad P_K^{(i)} = \{(j,k) \in \mathcal{E} : r_{jk}^{(i)} \leq K\}, \tag{31}$$

where $\mathcal{E}$ is the set of all 190 candidate undirected pairs.

To quantify agreement with the cohort-level reference, we treat the globally significant edge set $E_{\mathrm{GT}}$ (with $|E_{\mathrm{GT}}| = 41$) as ground-truth labels $y_{jk} = \mathbb{I}\{(j,k) \in E_{\mathrm{GT}}\}$, and binarize the sample-level top-$K$ prediction as $\hat{y}_{jk}^{(i)} = \mathbb{I}\{(j,k) \in P_K^{(i)}\}$. We then compute a per-sample error rate as the average of miss and false-alarm terms:

$$\mathrm{Err@}K(i) = \frac{1}{2}\left( \frac{|E_{\mathrm{GT}} \setminus P_K^{(i)}|}{|E_{\mathrm{GT}}|} + \frac{|P_K^{(i)} \setminus E_{\mathrm{GT}}|}{190 - |E_{\mathrm{GT}}|} \right), \tag{32}$$

and report the test-set average $\mathrm{Err@}K = \frac{1}{n_{\mathrm{test}}} \sum_{i \in \mathcal{I}_{\mathrm{test}}} \mathrm{Err@}K(i)$. As a random baseline, we uniformly sample $K$ edges from the 190 candidates as positives; in expectation, the miss and false-alarm terms are symmetric around $K/190$, yielding $\mathbb{E}[\mathrm{Err@}K] = 0.5$. Empirically, iLoRA achieves $\mathrm{Err@}19$ well below this random baseline (see Table 2), indicating that its highest-weight sample-level edges are globally enriched for cohort-supported pairs.

## K. Limitations

iLoRA has several limitations. First, the graph branch operates on a selected subset of entities. In the IBD experiments, we use MaAsLin2 to select statistically significant taxa before latent graph inference. This improves tractability and reduces noise in sparse, zero-inflated microbiome profiles, but it may miss informative long-tail taxa or interactions involving taxa excluded by the upstream feature-selection step. Second, the pairwise graph construction scales as $O(K^2)$ in the number of selected entities. Although this cost is modest for $K = 20$ in our experiments, substantially larger graphs would require approximate edge screening, sparse candidate generation, or hierarchical graph construction. Third, iLoRA introduces additional inference cost over static LoRA because it performs sample-specific graph inference and hypernetwork-based LoRA generation. In our setting this overhead is moderate and remains much smaller than Deep Ensemble, but it should be considered in latency-sensitive deployments. Fourth, the biomedical evaluation focuses on UC vs. CD diagnosis because this is the endpoint consistently available across the aggregated public cohorts. Important clinical tasks such as treatment-response prediction, disease severity estimation, and longitudinal risk modeling remain future work. Finally, the inferred interaction graphs should be interpreted as predictive and statistical interaction structures rather than causal microbial mechanisms; biological conclusions require external validation.

