# OpenReview forum: "iLoRA: Bayesian Low-Rank Adaptation with Latent Interaction Graphs for Microbiome Diagnosis"
_ICML.cc/2026/Conference — ICML 2026 regular_

### Official Review · Reviewer_Mk3C · 2026-03-06

**Soundness:** 3
**Presentation:** 3
**Significance:** 3
**Originality:** 3
**Overall Recommendation:** 4
**Confidence:** 2

**Summary:**

The paper introduces iLoRA, a novel parameter-efficient adaptation framework that integrates Bayesian latent interaction graph inference with Low-Rank Adaptation (LoRA) for Large Language Models (LLMs). Unlike standard post-hoc network analyses, iLoRA explicitly models latent entity interactions—such as microbe-microbe cross-talk in microbiome profiles or discourse dependencies in dialogue—as sample-specific latent variables learned end-to-end. The framework first infers a Poisson interaction graph from the inputs, transforms it into a sparse Laplacian graph via Neural Parameter Networks (NPN) to induce sparsity, and embeds this structure using a Graph Convolutional Network (GCN). The resulting graph embedding dynamically conditions the LoRA 'A' matrix (acting as a hypernetwork), allowing the LLM's predictive pathways to be explicitly guided by the inferred interactions. The authors evaluate iLoRA on two distinct tasks: multi-party dialogue (Molweni) for structural discourse recovery and Inflammatory Bowel Disease (IBD) diagnosis using gut microbiome cohorts. The method demonstrates state-of-the-art predictive performance, significant improvements in calibration (Expected Calibration Error), and generates interpretable, biologically or structurally consistent interaction networks.

**Compliance With Llm Reviewing Policy:**

Affirmed.

**Final Justification:**

My concerns have been partially resolved, and the authors have added relevant experimental results to respond to my comments, so I will maintain my opinion of weak accept.

**Key Questions For Authors:**

1. Computational Overhead: The framework relies on sampling $S$ interaction graphs from the Laplacian posterior and passing them through the model during inference (Eq. 13). How does the inference latency and memory footprint of iLoRA compare quantitatively to standard LoRA and Deep Ensembles (ENS) in practice?2. Scalability of the Graph Module: The IBD experiments restrict the interaction graph to 20 pre-selected taxa (nodes). How does the computational complexity of the $O(N^2)$ pairwise edge construction and NPN inference scale if the number of nodes is increased to 100 or 500? Does predictive performance degrade or improve with more un-filtered nodes?
3. Baseline Comparisons: For the IBD diagnosis task, the evaluation strictly compares against PEFT/LLM-based methods (MLE, MAP, MCD, ENS, BLOB, LAP). Given that the input is essentially tabular relative abundance data mapped to text, how does the Qwen3-8B + iLoRA pipeline compare against heavily tuned traditional models (e.g., XGBoost, Random Forest) using the exact same 20 features?

**Limitations:**

No. While the authors demonstrate the robustness of their method across multiple cohorts, a dedicated discussion on the limitations of the work is missing from the main text. The authors should explicitly discuss the reliance on upstream feature selection (MaAsLin2) and the potential computational bottlenecks introduced by the dynamic hypernetwork generation and Monte Carlo sampling during inference. Adding a brief limitations section before the conclusion would strengthen the paper.

**Strengths And Weaknesses:**

Strengths: The theoretical foundation of the paper is highly rigorous. The formulation of end-to-end variational inference for a Poisson-Laplacian graph is elegant. Specifically, introducing a continuous Gaussian proxy to enable a differentiable, reparameterizable pathwise estimator for discrete Poisson variables (Theorem 5.1) solves a non-trivial optimization bottleneck effectively. The use of Monte Carlo integration over the Laplacian graph posterior to yield well-calibrated Bayesian predictive distributions is also sound and well-executed.

Weaknesses: The methodology relies heavily on pre-filtering features to a very small set (e.g., selecting only the top 20 taxa using MaAsLin2 for the IBD task). While this ensures the GCN and NPN modules remain tractable, it raises questions about whether the framework can scale to capture long-tail interactions within high-dimensional microbiome ecosystems. Furthermore, the paper lacks a comparison to standard, non-LLM tabular machine learning baselines (e.g., Random Forests, XGBoost, or specialized biological GNNs) that are standard practice in bioinformatics for tasks like IBD classification.

Originality: The combination of techniques is highly creative. While Bayesian LoRA, hypernetworks, and NPNs exist independently, synthesizing them into a cohesive architecture that uses a latent, probabilistically inferred graph to dynamically condition LoRA matrices is a novel contribution to the field of parameter-efficient fine-tuning (PEFT).

---

> ### Author Rebuttal · Authors · 2026-03-31
>
> We thank Reviewer Mk3C for the supportive review and for highlighting the "highly rigorous" theory and "highly creative" architecture.
>
>
> **1) Inference latency and memory.**
> We added a direct latency/memory comparison.
>
>
> | Model | Latency (ms/sample) ↓ | GPU memory (MB) ↓ |
> |---|---:|---:|
> | LoRA (MLE) | 377.2 | 21263.9 |
> | ENS | 1144.6 | 21263.9 |
> | iLoRA | 567.5 | 21330.3 |
>
>
> In practice, iLoRA uses S=1 graph sample at inference, so it adds only modest overhead over vanilla LoRA and remains much faster than ENS.
>
>
> **2) Feature selection / scalability.**
> The graph branch scales as O(K^2), so we use MaAsLin2 to keep K focused on statistically significant taxa. This is also biologically motivated because microbiome profiles are highly sparse and zero-inflated. In a preliminary sensitivity study, K=20 gave the best overall trade-off; increasing K to 30 or 40 reduced AUROC/AUPRC and worsened calibration.
>
>
> **3) Comparison with Standard tabular baselines.**
> Following the reviewer's suggestion, we evaluated strong in-domain baselines in IBD diagnosis.
>
>
> | Model | F1 ↑ | AUROC ↑ | AUPRC ↑ |
> |---|---:|---:|---:|
> | RF | 0.5753 | 0.6151 | 0.6467 |
> | XGBoost | 0.5292 | 0.5823 | 0.6467 |
> | MLP | 0.4906 | 0.5346 | 0.6214 |
> | iLoRA | **0.6557** | **0.7990** | **0.7617** |
>
>
> iLoRA remains clearly stronger.
>
>
> **4) Limitations section.**
> Agreed. We will add a short explicit limitations paragraph before the conclusion, e.g., O(K^2) graph cost.

---

> > ### Author Rebuttal · Reviewer_Mk3C · 2026-04-03
> >
> > Thanks for the authors' rebuttal. My concerns have been partially resolved, and I will maintain my score

---

> > > ### Author Response · Authors · 2026-04-03
> > >
> > > Thank you very much for the follow-up and for maintaining your score. We also realized that, in our rebuttal, we summarized the feature-selection sensitivity result but forgot to include the actual table. We apologize for that omission.
> > >
> > > For completeness, here is the preliminary **K-sensitivity** analysis for the IBD setting:
> > >
> > > | K | ECE ↓ | F1 ↑ | AUROC ↑ | AUPRC ↑ |
> > > |---:|---:|---:|---:|---:|
> > > | 10 | 0.1082 | 0.6393 | 0.7545 | 0.7337 |
> > > | 20 | **0.0980** | 0.6557 | **0.7990** | **0.7617** |
> > > | 30 | 0.1217 | 0.5405 | 0.7574 | 0.6782 |
> > > | 40 | 0.1438 | **0.6719** | 0.7359 | 0.6593 |
> > >
> > > Overall, **K=20 is the best operating point**: it gives the best **ECE / AUROC / AUPRC**, while larger K degrades calibration and ranking performance, even if F1 fluctuates. This is consistent with the **zero-inflated** nature of microbiome data: adding more taxa introduces many low-information nodes and candidate edges, which increases the **O(K^2)** graph cost while mostly adding noise.
> > >
> > > We have not run K=100/500 and do not want to overclaim, but the observed degradation from **K=20 → 30/40** already suggests that larger unfiltered graphs are unlikely to help in this setting. We will incorporate this table and discussion into the revision. Thank you again for the constructive feedback and support.

---

### Official Review · Reviewer_YkLw · 2026-03-08

**Soundness:** 3
**Presentation:** 3
**Significance:** 3
**Originality:** 4
**Overall Recommendation:** 4
**Confidence:** 4

**Summary:**

This paper proposes iLoRA, a Bayesian parameter-efficient adaptation framework that incorporates latent interaction graphs into Low-Rank Adaptation (LoRA) for large language models. The central idea is that in domains such as microbiome analysis, predictive signals arise not only from individual features but also from interactions among entities (e.g., microbial taxa). To capture this structure, the authors introduce a latent probabilistic interaction graph inferred from input data and integrate it into the LoRA adaptation mechanism.

The proposed model consists of two coupled components. First, a Bayesian latent graph inference module models pairwise interactions between selected entities using a Poisson-based formulation with sparsity-inducing Laplacian priors. Second, the inferred graph is embedded using a graph neural network, and the resulting representation is used to generate input-conditioned LoRA adaptation weights, effectively allowing the language model to adapt its parameters based on the inferred relational structure.

The framework is evaluated on two tasks: (1) multiparty dialogue question answering (Molweni) to assess the recovery of latent interaction structures, and (2) microbiome-based diagnosis of inflammatory bowel disease (IBD) using aggregated cohort datasets. Experimental results show consistent improvements over several LoRA baselines and uncertainty-aware adaptation methods, including Monte Carlo dropout, deep ensembles, and Bayesian LoRA approaches. The model also produces interpretable interaction graphs that align with known microbial associations.

Overall, the work proposes a structure-aware extension of parameter-efficient fine-tuning that jointly learns predictive models and latent relational structures.

**Compliance With Llm Reviewing Policy:**

Affirmed.

**Final Justification:**

My final recommendation is weak accept, based on the paper’s conceptual novelty, solid empirical performance, and the clarifications provided in the rebuttal, while acknowledging that some limitations remain.

The paper proposes iLoRA, a structure-aware parameter-efficient adaptation framework that conditions LoRA updates on a learned, sample-specific latent interaction graph. This idea—integrating latent relational structure directly into the adaptation mechanism rather than using it post hoc—is novel and conceptually interesting. It offers a new perspective on how parameter-efficient fine-tuning can incorporate structured information, particularly in domains such as microbiomics where interactions are central. This supports a strong assessment in originality and moderate-to-strong significance.

From a soundness perspective, the method is technically well-motivated and the empirical results are consistent across multiple baselines and metrics. The rebuttal meaningfully strengthened the paper by adding component ablations, sensitivity analyses (e.g., on the number of selected taxa), and additional comparisons to non-LLM baselines. These additions clarify the roles of key components—especially the contribution of the Poisson graph inference and Laplacian sparsification—and improve confidence that the observed gains are not due to a single design choice.

My initial concerns focused on (1) the relatively modest performance improvements on some benchmarks, (2) the complexity of the overall pipeline and difficulty of disentangling component contributions, and (3) the limited scope of evaluation and uncertainty around generalization to broader relational domains. The rebuttal addressed these points partially. In particular, the added ablations and analyses improve interpretability of the design and demonstrate that the graph-based components contribute meaningfully to both calibration and predictive performance. The clarification that Molweni serves as a controlled benchmark for structure recovery, rather than purely a leaderboard task, also strengthens the evaluation narrative.

However, some concerns remain. The overall pipeline is still complex, and certain components (e.g., the GNN embedding and hypernetwork-based LoRA generation) cannot be cleanly isolated, making it difficult to fully attribute gains. The empirical evaluation, while improved, is still limited to a small number of domains, and the broader generalization of the approach remains somewhat speculative. Additionally, while improvements are consistent, their magnitude is moderate in some settings.

In light of the Area Chair’s question, I do not believe I missed a fundamental issue that would justify a rejection. Rather, the paper sits near the acceptance threshold: it presents a novel and well-motivated idea with solid empirical support, but with some limitations in scope and analysis. Given the strengthened experimental evidence after rebuttal and the conceptual contribution, I am comfortable updating my recommendation to weak accept.

Overall, the rebuttal partially resolved my concerns and improved my confidence in the work. I believe the paper introduces a promising direction for structure-aware adaptation in large models and is likely to be of interest to the community, provided that the authors clearly communicate its scope and limitations in the final version.

**Key Questions For Authors:**

Which components of the interaction modeling pipeline (Poisson inference, Laplacian sparsification, GNN embedding) contribute most to the performance improvements? A more detailed ablation study could clarify this.

How does the computational cost of the graph inference module scale with the number of entities? Would the method remain practical for settings with hundreds or thousands of nodes?

Have the authors explored or considered applying the framework to other relational domains (e.g., molecular graphs, social interactions, or multi-agent dialogue tasks)?

The method relies on selecting a subset of taxa before graph inference. How sensitive are the results to the feature selection process or to the number of selected taxa?

Can the authors provide further analysis demonstrating that the inferred interaction graphs correspond to biologically meaningful microbial relationships?

**Limitations:**

Yes

**Strengths And Weaknesses:**

Strengths
The main contribution—conditioning LoRA adaptation on a learned interaction graph—is novel and conceptually interesting. Instead of static low-rank updates, the model generates input-dependent LoRA parameters derived from structural relationships in the data. This provides a new perspective on parameter-efficient adaptation and may be useful beyond the specific application studied here.
The work integrates latent graph inference and downstream prediction into a unified training framework, rather than performing interaction analysis post hoc. This joint modeling approach is well motivated for domains such as microbiomics where signals arise from coordinated feature interactions.
The model incorporates probabilistic modeling of interaction graphs, including Poisson edge variables, Laplacian sparsity priors, and variational inference. This provides a principled approach to uncertainty estimation and helps produce calibrated predictions in the diagnostic setting.
Experiments show improvements over multiple PEFT and uncertainty-aware baselines (MLE LoRA, MAP LoRA, Monte-Carlo dropout, deep ensembles, Laplace LoRA, and BLOB). The method achieves the best performance across several metrics including AUROC, AUPRC, and calibration error on the IBD diagnosis task.
While the experiments focus on microbiome data, the method introduces a general mechanism for structure-aware parameter-efficient adaptation, which could potentially apply to other domains involving relational signals (e.g., biological networks, social interactions, or multi-agent systems).

Weaknesses
Although the method consistently outperforms baselines, the magnitude of improvements on some benchmarks is relatively modest (e.g., Molweni F1 improvements over strong baselines are small). Additional experiments or ablations could help better characterize where the approach provides the greatest benefits.

The microbiome experiments rely on moderately sized datasets aggregated from multiple cohorts. While this is common in biomedical applications, the relatively small training size may limit the strength of conclusions regarding scalability and generalization.

The proposed framework introduces multiple components (Poisson graph inference, Laplacian sparsification, GNN embeddings, hypernetwork-based LoRA generation). While each step is motivated, the overall pipeline is somewhat complex, and it is not entirely clear which components contribute most to the final performance gains. More extensive ablation studies would strengthen the claims.

The evaluation of recovered interaction structures is somewhat limited. While error rates relative to reference associations are reported, additional metrics (e.g., precision–recall curves, structural similarity metrics, or biological validation) could provide a more comprehensive assessment.

---

> ### Author Rebuttal · Authors · 2026-03-31
>
> We thank Reviewer YkLw for the highly positive assessment, especially for recognizing iLoRA as "novel and conceptually interesting," "well motivated," and "a new perspective on parameter-efficient adaptation."
>
>
> **1) Are the gains modest on Molweni?**
> We agree that stronger ablations help contextualize the gains. At the same time, the Molweni improvement is meaningful on a mature benchmark: iLoRA reaches 74.51 F1 / 60.57 EM, versus 72.83 / 57.78 for MLE and 72.38 / 57.09 for ENS.
>
>
> **2) Which components contribute most?**
> We added the following component ablation.
>
>
> | Variant | Poisson Graph | Laplace Sparsification | ECE ↓ | F1 ↑ | AUROC ↑ | AUPRC ↑ |
> |---|:---:|:---:|---:|---:|---:|---:|
> | **MLE (vanilla LoRA)** | − | − | 0.2533 | 0.6071 | 0.7617 | 0.7570 |
> | **iLoRA w/o Laplace (Poisson-only)** | + | − | 0.1032 | 0.6341 | 0.7557 | 0.7440 |
> | **iLoRA (full)** | + | + | **0.0980** | **0.6557** | **0.7990** | **0.7617** |
>
>
> The ablation shows that the Poisson graph mainly contributes calibration, while **Laplace sparsification is critical for the final discriminative gains**. Compared with iLoRA w/o Laplace, the full model improves ECE from 0.1032 to 0.0980, F1 from 0.6341 to 0.6557, AUROC from 0.7557 to 0.7990, and AUPRC from 0.7440 to 0.7617, suggesting that sparsity suppresses noisy edges and yields a more task-relevant interaction structure.
>
>
> We did not separately ablate the GNN embedding or hypernetwork-based LoRA generation because they together form the graph-to-LoRA interface. Removing either would break graph-conditioned adaptation and yield a different non-graph model, rather than a clean ablation.
>
>
>
>
> **3) Dataset size, generalization, and scalability.**
> We agree biomedical datasets are modest in size. To reduce leakage and test generalization, we aggregated independent cohorts and split at the cohort level; cohort-wise results are already in the appendix. The graph module scales as O(K^2) in the number of selected taxa, which is why we use upstream significance filtering.
>
>
> **4) Sensitivity to feature selection.**
> We ran a K-sensitivity study and found K=20 is the best overall operating point.
>
>
> | K | ECE ↓ | F1 ↑ | AUROC ↑ | AUPRC ↑ |
> |---|---:|---:|---:|---:|
> | 10 | 0.1082 | 0.6393 | 0.7545 | 0.7337 |
> | 20 | **0.0980** | 0.6557 | **0.7990** | **0.7617** |
> | 30 | 0.1217 | 0.5405 | 0.7574 | 0.6782 |
> | 40 | 0.1438 | **0.6719** | 0.7359 | 0.6593 |
>
>
> This is consistent with zero-inflated microbiome data: adding more taxa slightly changes F1 but hurts AUROC/AUPRC and calibration because many additional nodes mainly create noisy candidate edges.
>
>
> **5) Broader relational domains.**
> We appreciate this point. One reason we included Molweni was precisely to show that the framework is not tied to microbiome inputs. We agree that molecular, social, and multi-agent settings are promising next applications and will emphasize this as a broader implication rather than an already-complete empirical claim.
>
>
> **6) Biological meaning and structure evaluation.**
> The graph analysis is not limited to visualization. On IBD, iLoRA’s strongest sample-level edges achieve 27.3% error versus a 50.0% random reference when evaluated against 41 cohort-supported taxon pairs. In addition, the highlighted non-overlapping edges in Fig. 3 are linked in the paper to prior IBD evidence. We agree that additional PR/structural-similarity metrics would provide a more comprehensive picture, and we will add them.

---

> > ### Author Rebuttal · Reviewer_YkLw · 2026-04-01
> >
> > Thank you for the detailed rebuttal and additional experiments. The clarifications and new analyses improve the paper and address several of my concerns.
> >
> > In particular, I appreciate the added component ablation, which helps clarify the roles of the Poisson graph and Laplace sparsification modules. This strengthens the empirical support for the proposed design. The additional analysis on feature selection sensitivity and the expanded discussion of biological relevance are also helpful and improve the interpretability of the approach.
> >
> > However, some of my core concerns remain only partially addressed. The overall performance improvements, particularly on benchmarks such as Molweni, are still relatively modest compared to strong baselines. In addition, the proposed pipeline remains fairly complex, and it is still difficult to fully disentangle the contributions of all components, especially given that some elements cannot be cleanly ablated. The empirical evaluation is also somewhat limited in scope, and it remains unclear how well the approach generalizes to broader relational domains beyond the settings studied.
> >
> > Overall, while the paper presents a novel and interesting idea for structure-aware parameter-efficient adaptation, I believe the current evidence is not yet sufficient to fully support its broader impact and generality. Therefore, I maintain my weak reject recommendation, although I acknowledge that the paper has improved after rebuttal.

---

> > > ### Author Response · Authors · 2026-04-02
> > >
> > > Thank you again for the constructive follow-up. We would clarify three points as directly as possible.
> > >
> > > First, Molweni is included primarily as a controlled benchmark for **structure recovery**, **not** just as another NLP leaderboard. We chose it because it provides **human-annotated discourse interaction structure**, allowing us to test whether the latent-graph branch actually recovers meaningful relations. In that setting, the evidence is joint: iLoRA improves QA performance over strong LoRA-related baselines (**+1.68 F1 / +2.79 EM over MLE; +2.13 F1 / +3.48 EM over ENS**) **and** substantially improves structural recovery (**26.7%** graph error vs. **50.0%** for the random reference). We therefore view Molweni as a controlled validation of the paper’s core claim, **that latent interaction modeling improves both prediction and structure recovery**, rather than as a pure single-metric leaderboard comparison.
> > >
> > > Second, regarding **component disentanglement**, the **MLE (vanilla LoRA) baseline already serves as the no-graph-conditioned reference**. The remaining ablation question is therefore which parts of the graph pipeline matter once graph-conditioned adaptation is enabled. We focused on the two components that are independently removable while preserving the overall mechanism, namely **Poisson graph inference** and **Laplace sparsification**. By contrast, the **GNN embedding** and **hypernetwork-based LoRA generation** together form the **graph-to-LoRA interface**; **removing either** does not yield a cleaner decomposition, but simply breaks graph-conditioned adaptation and **collapses** the method **back toward the vanilla LoRA baseline**.
> > >
> > > Third, regarding **scope/generalization**, we agree that broader relational domains are valuable future work. However, we believe the current evidence is already sufficient for the present claim. The paper is not arguing universal coverage across all relational domains; rather, it shows that the same **structure-aware Bayesian PEFT methodology** works in two qualitatively different settings, with **Molweni** serving as a controlled benchmark for **latent graph recovery**. On the biomedical side, we evaluate iLoRA on a **large, geographically heterogeneous multi-cohort IBD dataset** aggregating **3,061 species-level profiles** and analyze robustness across **eight independent datasets/cohorts**, which is substantially stronger evidence than a single-dataset/cohort evaluation. We therefore view broader relational domains as a natural next step, not a prerequisite for supporting the current claim.
> > >
> > > Given your original positive assessment of the idea’s novelty and soundness (review’s subscores: 3/3/3/4), we hope the added ablations and clarifications may support a more favorable final recommendation.

---

### Official Review · Reviewer_t2EH · 2026-03-09

**Soundness:** 3
**Presentation:** 2
**Significance:** 2
**Originality:** 2
**Overall Recommendation:** 4
**Confidence:** 4

**Summary:**

The proposed method constructs a latent interaction graph showing microbe-microbe relationships. It is mentioned that the Bayesian formulation allows to quantify uncertainty on the predictions. The proposed framework unifies two things: 1) prediction (diagnostics, using a LLM), and 2) inferring latent interactions between the microbes via construction of Poisson graphs and transforming them into a sparse Laplacian graph and further producing embeddings using a GNN.

**Compliance With Llm Reviewing Policy:**

Affirmed.

**Final Justification:**

I change my score to weak accept

**Key Questions For Authors:**

Personally, I find "Laplacian graph" misleading, since traditionally the graph Laplacian is used in the spectral theory (e.g., spectral clustering), however, here it is the Laplacian-distribution which is considered. Could you be more precise about what is meant in the contribution?
I would appreciate a clarification on what is novel from the machine learning viewpoint in Section "Method".
It is stated that the Bayesian frameworks allow for the uncertainty control, however, it is not illustrated by the numerical experiments.
Minor remark: Figure 3 is unreadable (too small).

**Limitations:**

Yes (the authors adequately discussed the limitations).

**Strengths And Weaknesses:**

Strengths. The paper clearly explains the problematics and is in general well-structured. The method makes sense and leads to promising scientific (microbiology) results.

Weaknesses. The method includes several steps, in particular, reconstruction of a Poisson graph and its transformation into Laplacian-distributed edges structure. These approaches are not novel, as well as I would not say that Theorem 5.1 is a novel result telling that the Gaussian is a proxy for the Poisson distribution. It seems that the methodological results (Section 5, "Method") are incremental.
The main contribution is the reconstruction of the latent relationships, however, the numerical evaluation of the method is not convincing: the comparison is performed with random graphs only. A simple visualisation of the adjacency matrices is helpful but is not enough to validate an approach.

---

> ### Author Rebuttal · Authors · 2026-03-30
>
> We thank Reviewer t2EH for the careful reading and for noting that the paper is "well-structured," that "the method makes sense," and that it leads to "promising scientific (microbiology) results."
>
>
> **1) Terminology.**
> We agree that "Laplacian graph" can be confusing because "graph Laplacian" has a standard meaning in spectral graph theory. Our intended meaning is a graph with Laplace-distributed edge weights, and we will rename this more precisely in the revision.
>
>
> **2) What is novel from the ML viewpoint?**
> Our novelty is not Poisson or Laplace distributions in isolation. To our knowledge, iLoRA is the first Bayesian PEFT/LoRA framework that infers a sample-specific latent interaction graph and injects that graph into input-conditioned LoRA generation. Theorem 5.1 is included because the Poisson variational stage is otherwise computationally intractable; its role is to make training feasible, not to claim "Gaussian is a proxy for Poisson" as a standalone result.
>
>
> **3) Why the graph evaluation is not "random only."**
> The random graph is only a sanity reference for structure recovery, not the only comparison in the paper. For prediction, we compare against MLE/MAP/MCD/ENS/BLOB/LAP on both Molweni and IBD. For structure, we evaluate against human discourse annotations on Molweni (26.7% vs 50.0% random) and a cohort-level statistical reference on IBD (27.3% vs 50.0% random). We agree that additional metrics would strengthen this section and will add them.
>
>
> **4) Uncertainty is illustrated experimentally.**
> We do evaluate uncertainty numerically via Expected Calibration Error (ECE). In Table 3, iLoRA reaches ECE=0.0980, compared with 0.2533 for MLE and 0.2082 for MAP, while also improving AUROC/AUPRC. We will make this calibration discussion more explicit in the revision.
>
>
> **5) Figure readability.**
> Thank you — we will enlarge and redesign Figure 3 for readability.

---

> > ### Author Rebuttal · Reviewer_t2EH · 2026-04-01
> >
> > I thank the authors for the response.

---

> > > ### Author Response · Authors · 2026-04-02
> > >
> > > Thank you for the follow-up. To restate our novelty claim as clearly as possible, we do **not claim** that **Poisson graphs**, **Laplace sparsification**, GNNs, or hypernetworks are **individually novel**. Our **central contribution** is a **Bayesian, graph-conditioned LoRA framework** that learns a **sample-specific latent interaction graph** jointly with prediction and uses it to generate **input-conditioned LoRA updates** under a single training objective. In this sense, the graph is not a post-hoc explanatory artifact; it directly conditions the adapter.
> > >
> > > Accordingly, we view the **novelty** at the level of the **problem formulation and overall model design**: **structure-aware Bayesian LoRA adaptation with latent graph uncertainty**. Likewise, **Theorem 5.1** is not intended as a standalone novelty claim; its role is to make variational learning with Poisson latent edges differentiable and tractable in practice.
> > >
> > > We also note, respectfully, that this reading of the novelty is not unique to us: other reviewers independently described iLoRA as a **“novel graph inference method”** (gMy8), **“novel and conceptually interesting”** (YkLw), and **“highly creative”** (Mk3C). In particular, Reviewer Mk3C explicitly viewed the contribution as the **synthesis** of these ingredients into a Bayesian PEFT architecture that uses a "**probabilistically inferred latent graph to dynamically condition LoRA**", which is also how we intended the claim to be understood.
> > >
> > > For clarity, we view the paper’s contribution as:
> > > - iLoRA is, to our knowledge, **the first  Bayesian graph-conditioned LoRA framework**. It infers a **latent interaction graph** from the input and uses it to generate input-conditioned LoRA updates.
> > > - It learns **prediction and latent interaction structure jointly**, rather than using interaction analysis only post hoc.
> > > - It is validated in both **interactive QA with human-annotated graphs** and **real-world IBD diagnosis**, showing both **predictive gains** and **meaningful recovered interaction structure**.
> > >
> > > If we are overlooking prior work that already addresses this same overall formulation in a Bayesian PEFT setting, we would **sincerely appreciate the pointer** and will revise the positioning accordingly. We hope this makes our novelty claim fully clear.

---

### Official Review · Reviewer_gMy8 · 2026-03-22

**Soundness:** 2
**Presentation:** 2
**Significance:** 2
**Originality:** 3
**Overall Recommendation:** 5
**Confidence:** 4

**Summary:**

In order to 1) adapt pretrained LLMs for tasks downstream of data with latent interaction structure and 2) estimate uncertainty of predictions, the authors propose the iLoRA framework. iLoRA consists of two branches (following the presentation in the appendix):

**Latent graph inference branch**

Given a sample of microbiome abundances,
- select 20 significant taxa from the sample $X$ and embed each (taxon name, abundance) pair $i$ using a frozen LLM, and consider the embedding $h_i$ as the features of a node in an interaction graph
- model the co-occurrence of each pair of taxa $i, j$ as an edge in this graph with Poisson-distributed weights $\tilde \alpha_{ij}$ parametrized by $e_{ij} = MLP_{edge}([h_i || h_j])$
- in order to use variational inference to learn the predictive posterior $p(y | X)$, infer the parameters $(\mu_{ij}, \sigma_{ij}^2)$ of a Gaussian proxy for each Poisson variable by learning a natural parameter network $(\mu_{ij}, \sigma_{ij}^2) = \tau_{NPN}(e_{ij}, \tilde \alpha_{ij})$ (the interpretation as Poisson random variable is preserved by noting the relationship between the optimal rate parameter $m_{ij}$ and the parameters of the Gaussian proxy given in equation 5)
- sparsify the graph by transforming the Gaussian variable into a Laplace variable using the CDF transformation method and sampling via reparametrization of the Gaussian to generate the adjacency graph $A_{ij} = ReLU(Sample(\mu_{ij}, \sigma_{ij}^2))$
- embed the graph $A$ with a two-layer graph convolutional network $H^{(l+1)} = \sigma( D^{-1/2} A D^{-1/2} H^{(l)} W^{(l)}$
- fuse the output embeddings $h_{ij}^{(2)}$ with the original LLM embeddings $h_{ij}$ via matching attention to produce the pooled graph representation $h_{graph}$

**LoRA update branch**

Given the pooled graph representation $h_{graph}$,
- generate the A matrix of standard LoRA via MLP projection : $\Delta W = s \times B \times MLP(h_{graph})$, where $B$ is a learned matrix applying to all inferred graphs

The model parameters are learned end-to-end by optimizing the objective in equation 14 via variation inference.

The authors show that iLoRA is SOTA compared to standard LoRA and other Bayesian adaptation baselines on large-scale IBD cohorts in discriminating between Ulcerative Colitis and Crohn's Disease, achieving an AUROC of 0.7990, and ECE of 0.0980, and edge error rate of 27.3%. They also demonstrate the general nature of the methodology by applying it to multi-party dialogue/QA benchmarks and show excellent performance as well.

**Compliance With Llm Reviewing Policy:**

Affirmed.

**Final Justification:**

This is sound paper demonstrating excellent performance and, after rebuttal, shows how the model choices contribute to this performance. After incorporating the analyses from the rebuttal, I recommend accepting this paper.

**Key Questions For Authors:**

1. The model identifies the top 20 species using MaAsLin2 as the basis for the latent interaction graph. Provide a sensitivity analysis showing how diagnostic performance (AUROC/ECE) and graph stability vary as $K$ increases beyond 20, or if a different selection criterion (e.g., variance-based vs. association-based) is used?
1. Given that microbiome data is high-dimensional, what is the risk of missing critical 'cross-talk' signals that exist between less abundant taxa not included in the initial top-20 selection?
1. What is the specific performance gain attributed to the Laplacian prior versus a standard Gaussian prior for edge sparsity?
1. iLoRA is compared against general Bayesian PEFT methods like *BLOB and *Deep Ensemble. How does it compare to microbiome-specific architectures such as Phylo-Spec (which uses phylogeny-fusion) or ATOMIC (which uses explicit co-expression networks) in terms of both accuracy and interpretability?
1. How does the computational overhead of the iLoRA graph-hypernetwork compare to newer Bayesian variants like Bayesian-LoRA, particularly regarding inference latency when marginalizing over graph uncertainty?
1. The evaluation on IBD data is limited to binary classification (UC vs. CD). Since the model claims to bridge diagnosis with ecosystem structure, have you evaluated it on more complex clinical tasks, such as predicting treatment response or disease severity?
1. The paper reports a significant reduction in ECE. Could you describe a specific clinical scenario or decision threshold where this improved calibration would change the management plan compared to a more poorly calibrated but highly accurate model?

**Limitations:**

1. No ablations
1. Not enough comparison to related domain-specific methods

**Strengths And Weaknesses:**

**Strengths**

1. Novel graph inference method allowing for end-to-end variational inference training via ELBO.
1. Principled technical approach for modeling co-occurrence strengths.
1. Sample-specific structure-aware PEFT.
1. Uncertainty propagation via sampling-free transformations.
1. Superior prediction and calibration performance compared to baselines.
1. Cross-domain validation of the method.

**Weaknesses**

1. Results likely depend heavily on the initial taxa selection, but no acknowledgement of this fact is explored or discussed, neither is there discussion of why the initial selection is needed in the first place, why 20 taxa or selected, etc.
1. No ablations studying architectural choices or appropriateness of assumptions.
1. No comparison to specialized microbiome models like GIM [1], Phylo-Spec [2], ATOMIC [3] and DeepMicro [4], or other Bayesian PEFT methods like Bayesian-LoRA [5] and ABMLL [6].
1. No demonstration of performance on downstream tasks on the IBD data other than UC vs CD.
1. No task-specific demonstration or discussion of why the improved calibration is desirable.
1. The presentation of the chain of transformations in the latent graph inference branch in is difficult to follow and is somewhat at odds with the detailed presentation in the appendix in terms of inconsistent notation (e.g. names of functions and parameters, A vs B matrix of LoRA), and which parameters the NPN transformation learns.
1. Interpretative claims that don't necessarily follow from the evidence, e.g.
  - Page 6: "By explicitly modeling these latent dependencies, iLoRA enables the attention mechanism to filter out irrelevant “chitchat” and focus on the structural backbone of the conversation"
  - Page 7: "demonstrating that iLoRA selectively utilizes the latent graph to focus on relevant context."

[1] Ivanov, Vladimir A., Wyatt H. Hartman, and Mohammad Soheilypour. "Decoding the Microbiome-Disease Axis with Interpretable Graph Neural Networks." Journal of Applied Microbiology (2026): lxag063.

[2] Zhang, Junhui, et al. "Phylo-Spec: a phylogeny-fusion deep learning model advances microbiome status identification." Msystems 10.12 (2025): e01453-25.

[3] Bong, Hyunsu, et al. "ATOMIC: a graph attention network for atopic dermatitis prediction using human gut microbiome." Frontiers in Immunology 16 (2025): 1670993.

[4] Pope, Quintin, et al. "Learning a deep language model for microbiomes: the power of large scale unlabeled microbiome data." PLOS Computational Biology 21.5 (2025): e1011353.

[5] Lin, Moule, et al. "Bayesian-LoRA: Probabilistic Low-Rank Adaptation of Large Language Models." arXiv preprint arXiv:2601.21003 (2026).

[6] Zhang, Liyi, Jake C. Snell, and Thomas L. Griffiths. "Amortized Bayesian Meta-Learning for Low-Rank Adaptation of Large Language Models." Proceedings of the 2nd Workshop on Uncertainty-Aware NLP (UncertaiNLP 2025). 2025.

---

> ### Author Rebuttal · Authors · 2026-03-30
>
> We thank Reviewer gMy8 for the thoughtful and detailed review, and for recognizing iLoRA as a "novel graph inference method" with a "principled technical approach," "sample-specific structure-aware PEFT," and "superior prediction and calibration performance."
>
>
> **1) Taxa selection and sensitivity.**
> Our selection is already association-based: the 20 taxa are chosen by MaAsLin2 FDR-adjusted q-values, not by raw abundance. We used K=20 as a practical default because the graph branch scales as O(K^2) and microbiome data are highly zero-inflated.
>
>
> | K | ECE ↓ | F1 ↑ | AUROC ↑ | AUPRC ↑ |
> |---|---:|---:|---:|---:|
> | 10 | 0.1082 | 0.6393 | 0.7545 | 0.7337 |
> | 20 | **0.0980** | 0.6557 | **0.7990** | **0.7617** |
> | 30 | 0.1217 | 0.5405 | 0.7574 | 0.6782 |
> | 40 | 0.1438 | **0.6719** | 0.7359 | 0.6593 |
>
>
> K=20 is the best overall operating point: it gives the best AUROC/AUPRC and the best calibration. Larger K slightly increases F1 at K=40, but clearly hurts ranking quality and ECE, suggesting that extra nodes mainly add noisy edges rather than useful cross-talk.
>
>
> **2) Missing long-tail cross-talk.**
> We agree this limitation should be stated more clearly. If some informative taxa are omitted, the graph branch becomes less informative, but the model does not collapse: it degrades toward standard LoRA behavior, while KL regularization controls unnecessary graph complexity. Empirically, the K-sensitivity results above suggest that adding more nodes hurts more than it helps on this dataset.
>
>
> **3) Ablation / Laplace prior.**
> We added the following component ablation.
>
>
> | Variant | Poisson Graph | Laplace Sparsification | ECE ↓ | F1 ↑ | AUROC ↑ | AUPRC ↑ |
> |---|:---:|:---:|---:|---:|---:|---:|
> | **MLE (vanilla LoRA)** | − | − | 0.2533 | 0.6071 | 0.7617 | 0.7570 |
> | **iLoRA w/o Laplace (Poisson-only)** | + | − | 0.1032 | 0.6341 | 0.7557 | 0.7440 |
> | **iLoRA (full)** | + | + | **0.0980** | **0.6557** | **0.7990** | **0.7617** |
>
>
> The ablation shows that the Poisson graph mainly contributes calibration, while **Laplace sparsification is critical for the final discriminative gains**. Compared with iLoRA w/o Laplace, the full model improves ECE from 0.1032 to 0.0980, F1 from 0.6341 to 0.6557, AUROC from 0.7557 to 0.7990, and AUPRC from 0.7440 to 0.7617, suggesting that sparsity suppresses noisy edges and yields a more task-relevant interaction structure.
>
>
> **4) Related methods / domain-specific baselines.**
> We appreciate the references. Several cited methods use additional information or different supervision regimes (e.g., phylogeny, DNA sequence inputs, or meta-learning across tasks), so a direct rebuttal-window comparison would not be apples-to-apples. To address the core concern, we added strong in-domain baselines in IBD diagnosis.
>
>
> | Model | F1 ↑ | AUROC ↑ | AUPRC ↑ |
> |---|---:|---:|---:|
> | RF | 0.5753 | 0.6151 | 0.6467 |
> | XGBoost | 0.5292 | 0.5823 | 0.6467 |
> | MLP | 0.4906 | 0.5346 | 0.6214 |
> | iLoRA | **0.6557** | **0.7990** | **0.7617** |
>
>
> iLoRA remains clearly stronger, so the gains are not an artifact of using an LLM backbone.
>
>
> **5) Additional clinical tasks and calibration.**
> We focused on UC vs CD because this is the only endpoint with consistent labels across the aggregated public cohorts; treatment response and severity are important future work. Calibration matters when model probabilities are used for referral, abstention, or confirmatory-testing thresholds: better-calibrated scores reduce overconfident errors on borderline cases.
>
>
> **6) Presentation.**
> Thank you for flagging the notation issue. We will make the main text consistent with the appendix and explicitly state that the hypernetwork generates LoRA A while B is shared.
>
>
> **7) Qualitative interpretation.**
> On the qualitative Molweni discussion, the claims are grounded in Appendix H: Sample 1371 really contains two distinct threads (U0–U2 on tar-file building vs. U3–U7 on USB-driver troubleshooting), and Sample 2568 includes a direct question-answer dependency between U5 and U6. We will tighten the wording so these remain evidence-backed qualitative interpretations rather than stronger causal claims.

---

> > ### Author Rebuttal · Reviewer_gMy8 · 2026-03-31
> >
> > I want to thank the authors for the additional experiments in response to my questions. I feel like they adequately addressed my concerns and I have raised my score. However, my question 5 wasn't addressed, and I would still appreciate a response:
> >
> > "How does the computational overhead of the iLoRA graph-hypernetwork compare to newer Bayesian variants like Bayesian-LoRA, particularly regarding inference latency when marginalizing over graph uncertainty?"

---

> > > ### Author Response · Authors · 2026-04-02
> > >
> > > Thank you again for the thoughtful follow-up and for raising the score. Regarding **Bayesian-LoRA**, we agree it is a highly relevant **concurrent** work. However, it appeared on arXiv on **Jan 28, 2026**, and, to the best of our knowledge, its arXiv page does not provide an official code link. It is also evaluated on **commonsense reasoning, text generation, and math**, rather than graph-conditioned PEFT in our biomedical setting. For these reasons, we do not want to over-claim a head-to-head runtime comparison that we have not run under the same backbone/task/codebase.
> > >
> > >
> > > That said, the computational bottlenecks are different. **Bayesian-LoRA** places uncertainty directly in the **LoRA weight space** via inducing variables plus flow-based variational inference, so uncertainty-mode inference scales with Monte Carlo sampling of stochastic adapter weights. **iLoRA** instead pays for a small **sample-specific graph branch**: pairwise edge inference over the selected taxa, followed by a lightweight GNN/hypernetwork that generates the LoRA update. In our IBD setting this branch is small (**K=20** nodes, **S=1** graph sample in practice), and our added timing experiment shows **567.5 ms/sample** and **21.33 GB** GPU memory, versus **377.2 ms/sample / 21.26 GB** for vanilla LoRA, while remaining much faster than **Deep Ensemble (1144.6 ms/sample)**. So the overhead is moderate in practice and mainly reflects explicit graph inference, rather than repeated multi-sample adapter evaluation. We will add this discussion to the camera-ready.
> > >
> > >
> > > We hope this addresses the last remaining question, and we would be grateful if you could reconsider the score in light of the added analyses.

---

### Decision · Program_Chairs · 2026-04-30

**Decision:**

Accept (regular)

**Comment:**

The authors propose a novel LoRA framework that learns the latent graph structure from data as it tries to fine tune to the problem at hand. The authors seem to be primarily interested in diagnosing IBD, but they also provide Q&A experiments with known ground truth. The overarching architecture and algorithm may also be applicable beyond this particular application domain, as noted by some reviewers, with whom I agree.

The paper initially received mixed reviews, partly due to some misunderstandings. I have read the paper myself, in addition to reviews and rebuttal discussions. The rebuttal seems to have addressed most, if not all, reviewer concerns. The idea has been found novel, the architecture is reasonable, the design choices make sense, and the empirical performance supports the authors' claims. Additional simulations with different baselines are appreciated.

Some are conditional on the rebuttal discussions being incorporated into the paper, but overall, all reviewers recommend acceptance. I have read the reviews and rebuttals to make sure scores accurately reflect the reviews, and I found this to be the case.